# Pressure-induced emission of cesium lead halide perovskite nanocrystals

Zhiwei Ma[1], Zhun Liu[2], Siyu Lu[3], Lingrui Wang[1], Xiaolei Feng [4,5], Dongwen Yang[2], Kai Wang[1], Guanjun Xiao[1], Lijun Zhang [1,2], Simon A.T. Redfern [4,5] & Bo Zou [1]

Metal halide perovskites (MHPs) are of great interest for optoelectronics because of their high quantum efficiency in solar cells and light-emitting devices. However, exploring an effective strategy to further improve their optical activities remains a considerable challenge. Here, we report that nanocrystals (NCs) of the initially nonfluorescent zero-dimensional (0D) cesium lead halide perovskite $Cs_4PbBr_6$ exhibit a distinct emission under a high pressure of 3.01 GPa. Subsequently, the emission intensity of $Cs_4PbBr_6$ NCs experiences a significant increase upon further compression. Joint experimental and theoretical analyses indicate that such pressure-induced emission (PIE) may be ascribed to the enhanced optical activity and the increased binding energy of self-trapped excitons upon compression. This phenomenon is a result of the large distortion of $[PbBr_6]^{4-}$ octahedral motifs resulting from a structural phase transition. Our findings demonstrate that high pressure can be a robust tool to boost the photoluminescence efficiency and provide insights into the relationship between the structure and optical properties of 0D MHPs under extreme conditions.

[1] State Key Laboratory of Superhard Materials, College of Physics, Jilin University, Changchun 130012, China. [2] Key Laboratory of Automobile Materials of MOE, and School of Materials Science, Jilin University, Changchun 130012, China. [3] College of Chemistry and Molecular Engineering, Zhengzhou University, Zhengzhou 450001, China. [4] Department of Earth Sciences, University of Cambridge, Downing Street, Cambridge CB2 3EQ, UK. [5] Center for High Pressure Science and Technology Advanced Research, Shanghai 201203, China. Correspondence and requests for materials should be addressed to G.X. (email: xguanjun@jlu.edu.cn) or to L.Z. (email: lijun_zhang@jlu.edu.cn) or to B.Z. (email: zoubo@jlu.edu.cn)

The recent success of organometallic halide perovskite nanocrystals (NCs) in photovoltaic devices has further triggered research activities on inorganic metal halide perovskite (MHP) NCs due to their good stability compared to their organic counterparts[1–6]. The cesium lead halide perovskite of $Cs_4PbBr_6$ is a typical zero-dimensional (0D) inorganic MHP, in which the octahedra are completely isolated by cation bridges and charge carriers are localized within the ordered metal halide component[7–10]. The electronic structure and optical properties of these materials are expected to show a significant dependence on pressure[11–16]. In view of this, our group has explored their pressure dependence and found, for example, that the band gap alignment of $CsPbBr_3$ NCs can be successfully fine-tuned via pressure[17]. In addition, Chen et al. demonstrated that pressure-sintered $CsPbBr_3$ nanoplatelets show a 1.6-fold enhancement in photoluminescence (PL) and display longer emission lifetimes than untreated NCs[13]. However, such studies have been largely limited to three-dimensional (3D) perovskites, without consideration of any low-dimensional network analogs. Furthermore, developing an effective strategy to improve the optical properties of MHPs remains a pressing challenge.

Here, we carried out a systematic high-pressure study of the NCs of a typical 0D perovskite, $Cs_4PbBr_6$. We found that the $Cs_4PbBr_6$ NCs undergo a structural phase transition from rhombohedral to monoclinic structure upon compression, accompanied with considerable structural distortion. First-principles energetic calculations indicate that the monoclinic structure is energetically more favorable than the rhombohedral phase with increasing pressure. The $Cs_4PbBr_6$ NCs exhibit an unexpected pressure-induced emission (PIE) at room temperature when the intrinsically nonemitting nanomaterials are compressed to 3.01 GPa. The underlying mechanism is attributed to the formation of self-trapped excitons (STEs). In particular, the room-temperature luminescence may be attributed to the enhanced optical activity and the increased binding energy of STEs in the high-pressure phase, which result from the large distortion and increased stiffness of $[PbBr_6]^{4-}$ octahedra upon compression. Our results suggest that pressure processing offers an exciting means to achieve perovskite materials that may show enhanced functional properties upon overcoming the limitations of conventional synthetic chemistry.

## Results and discussion

### Characterization of $Cs_4PbBr_6$ NCs under ambient conditions.

As shown in Fig. 1a, b, the samples before compression exhibit a well-defined morphology with good monodispersity. The $Cs_4PbBr_6$ NCs have an average diameter of 14.4 nm with a standard deviation of 1.3 nm. The lattice spacing of 0.68 nm measured from the HRTEM image (Fig. 1b) corresponds to the spacing of the (110) planes of rhombohedral $Cs_4PbBr_6$. Elemental mapping indicates that the elements of Cs, Pb, and Br are homogeneously distributed throughout the entire sample, which suggests high purity of the final products (Fig. 1c). Figure 1d, e illustrate the crystal structures of the 0D perovskite $Cs_4PbBr_6$ in its rhombohedral phase along and perpendicular to the c axis. Within the $Cs_4PbBr_6$ NCs, the $[PbBr_6]^{4-}$ octahedra are completely decoupled in all dimensions as a result of minimal electronic overlap between the adjacent octahedra. Therefore, the 0D perovskite NC system always exhibits a molecular-like absorption, and the size of the NCs does not have any remarkable effect on the band structure, which is consistent with the report by Manna[8,18]. Meanwhile, this unique structure renders a strong quantum confinement to confine charge carriers inside the octahedra to easily form bound excitons, which is exemplified by the clear and sharp excitonic peak in the absorption spectrum of

this material without applied pressure, as shown in Supplementary Fig. 1. We calculated the electronic band structure and density of states of $Cs_4PbBr_6$ at ambient pressure through first-principles calculations with spin-orbit coupling included (Supplementary Fig. 2). The separation of the conduction band minimum and the valence band maximum at the Z point of the Brillouin zone gives a direct band gap of ~3.23 eV. The calculated total and partial densities of states demonstrate that the valence bands primarily originate from the hybridization between the 6s orbital of Pb and the 4p orbital of Br, while the conduction band is mainly contributed by the 6p orbital of Pb. The steady-state PL of samples was obtained under ambient conditions, and no emission was observed, consistent with previous reports on bulk $Cs_4PbBr_6$ powders and films[19,20]. The absence of luminescence at room temperature of $Cs_4PbBr_6$ has been attributed to the thermal quenching effect[8,19] that may be caused by exciton migration[21], nonradiative recombination processes involving phonons[19], etc. Although some works showed that $Cs_4PbBr_6$ NCs or the bulk structure possess fluorescence in the visible region[22–24], an ongoing debate about the origin of the emission for these systems remains. The recent report by Manna et al.[8,18] proposed a more reasonable explanation, deepening the insight into the non-fluorescent mechanism of the $Cs_4PbBr_6$ systems. According to their reports[8,18], the origin of PL can be attributed to the presence of 3D perovskite impurities embedded inside the $Cs_4PbBr_6$ matrix, rather than to the intrinsic emission of $Cs_4PbBr_6$.

### In situ high-pressure optical measurement of $Cs_4PbBr_6$ NCs.

It has been previously suggested that the emission of perovskites might be greatly influenced by the nature of the octahedra[25]. Accordingly, intriguing PL properties might be expected as a result of structural modulation under varying pressure. For our sample, the pressure-dependent PL spectra were recorded up to 18.23 GPa (Fig. 2). We observe that $Cs_4PbBr_6$ NCs initially exhibit no PL response to the external pressure below 3.01 GPa. Above this value, a broad emission band with a full width at half maximum (FWHM) of ~150 nm suddenly appears (Fig. 2a). This drastic change in light emission is governed by the pressure-induced structural phase transformation (as demonstrated below). Upon further compression, the fluorescence of $Cs_4PbBr_6$ NCs exhibits an unambiguous pressure-sensitive evolution. Note that the corresponding PL intensity shows a remarkable increase with increasing pressure, eventually reaching a maximum at the pressure of 6.23 GPa (Fig. 2b and Supplementary Fig. 3a). The pressure dependences of the PL wavelength and the FWHM in our $Cs_4PbBr_6$ NCs are shown in Supplementary Fig. 3b, c. We can see that the PL peak of $Cs_4PbBr_6$ NCs shows an initial blueshift upon compression to 6.23 GPa. At higher pressure, it displays a redshift to 18.23 GPa, at which point the PL intensity almost disappears (Fig. 2c). The FWHM of the PL peak sharply decreases in the pressure region from 3.01 to 6.17 GPa, followed by a relatively sluggish increase.

The profile of the emission under high-pressure conditions appears asymmetric and skewed to the low-energy side, with a very large Stokes shift exceeding 190 nm. We attribute this to the formation of STEs. The radiative recombination of STEs is a well-known mechanism accounting for the large Stokes shifts observed in a number of broadly emitting lead halide perovskites, such as the recently reported two-dimensional (2D), one-dimensional (1D), and 0D systems[26–28]. An STE represents a photoinduced electron-hole pair (exciton) mediated by the interaction between the exciton and the corresponding lattice. Because of the photoinduced Jahn-Teller distortion, which causes substantial octahedral distortion upon photoexcitation, the STE usually acts as a state with a large Stokes shift. The formation of a localized

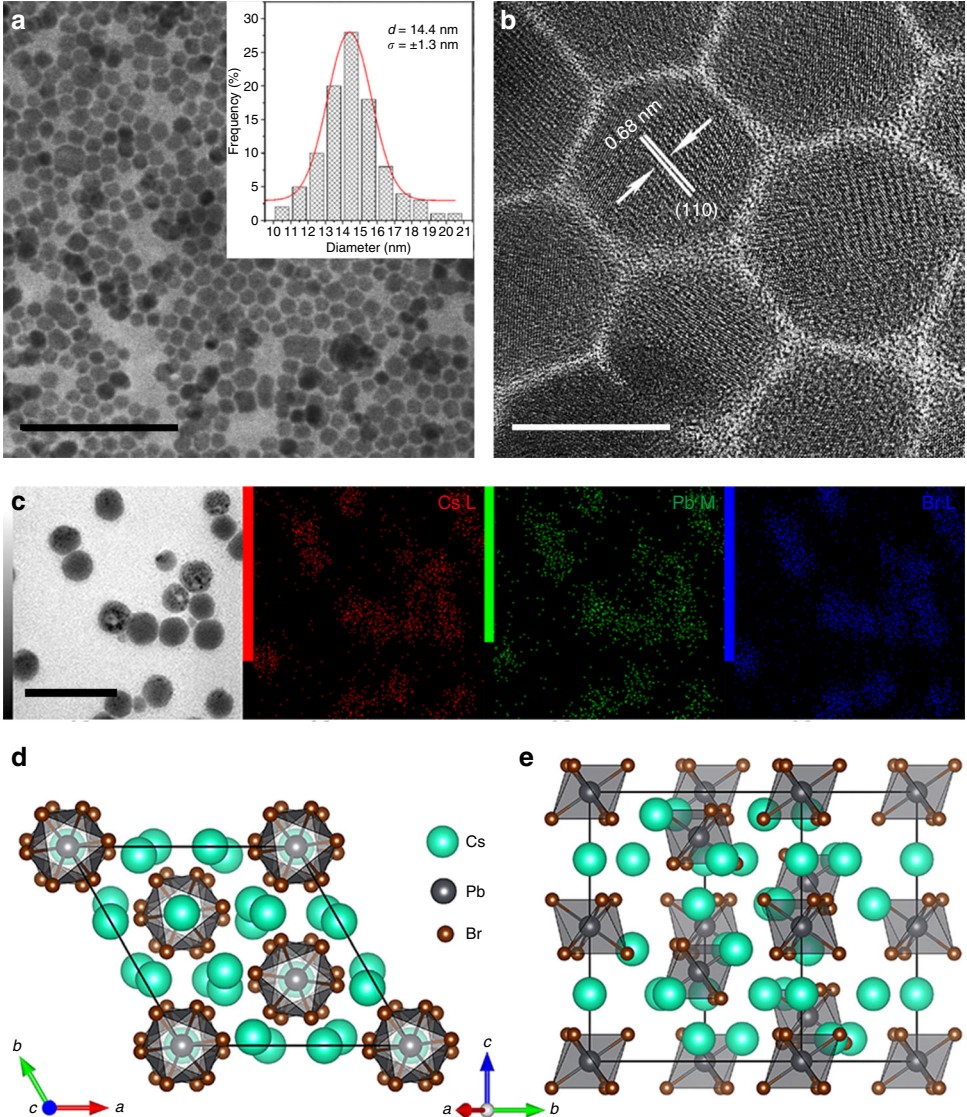

**Fig. 1** Structural characterization of $Cs_4PbBr_6$ NCs. **a** TEM image and the corresponding size distribution of as-prepared $Cs_4PbBr_6$ NCs with Gauss fitting, given in the inset. **b** High-resolution TEM (HRTEM) image of $Cs_4PbBr_6$ NCs before compression. Scale bars are 100 nm in the TEM image and 10 nm in the HRTEM image. **c** HAADF-STEM image and elemental mapping (Cs, Pb, and Br) of the as-prepared $Cs_4PbBr_6$ NCs (scale bar is 50 nm). Schematic crystal structure of $Cs_4PbBr_6$ along (**d**) and perpendicular (**e**) to (001)

STE is critically dependent on the dimensionality of the crystalline systems, and a more reduced dimensionality makes exciton self-trapping easier[29].

To better understand the pressure-induced emission in $Cs_4PbBr_6$ NCs, in situ high-pressure angle-dispersive synchrotron X-ray diffraction (ADXRD) patterns of the $Cs_4PbBr_6$ NCs were collected, as shown in Supplementary Figs. 4 and 5. The evolution of the ADXRD patterns demonstrates that a reversible structural phase transition from a rhombohedral (Phase I, space group $R$-$3c$) to a monoclinic (Phase II, space group $B2/b$) structure begins at 3.04 GPa and ends at 4.01 GPa (see Supplementary Figs. 4, 5, and 6 for more details). First-principles enthalpy calculations indicate that the monoclinic structure is energetically more favorable than the rhombohedral phase with increasing pressure (Supplementary Fig. 7). The phase transition is accompanied by a significant distortion of $[PbBr_6]^{4-}$ octahedra (Supplementary Fig. 8). In addition, the evolution of the Raman spectra ranging from 70 to 300 $cm^{-1}$ is shown in Supplementary Fig. 9. Three lattice modes at 77, 88, and 127 $cm^{-1}$ are observed under ambient conditions. These strong modes are associated with the

vibrational modes of $[PbBr_6]^{4-}$ octahedra[30]. As the pressure is increased to 3.10 GPa, the two lattice modes in the 70–100 $cm^{-1}$ region become very weak, while the relative intensity of the lattice mode at 127 $cm^{-1}$ is enhanced dramatically, consistent with the ADXRD result. The three modes undergo a redistribution of intensities and remain stable beyond 4.08 GPa, reflecting that the structure of the octahedra changed due to the phase transition. This phenomenon also agrees well with Rietveld refinement results. In this 0D perovskite material, the $Cs^+$ cations isolate each $[PbBr_6]^{4-}$ octahedra. Based on these results, the photoluminescent behavior that emerges cannot be explained as a result of lattice defects, but rather is due to the excited-state structural reorganization within individual $[PbBr_6]^{4-}$ octahedra[18,28]. Such excited-state structural reorganization corresponds to STE formation[28]. Herein, the role of high pressure is to drive the phase transition and increase structural distortion, which promotes the radiative recombination of STEs. Note that at the lower pressures where the rhombohedral phase dominates, compression makes the six bond lengths of Pb–Br within the regular $[PbBr_6]^{4-}$ octahedra shorter (Supplementary Fig. 10).

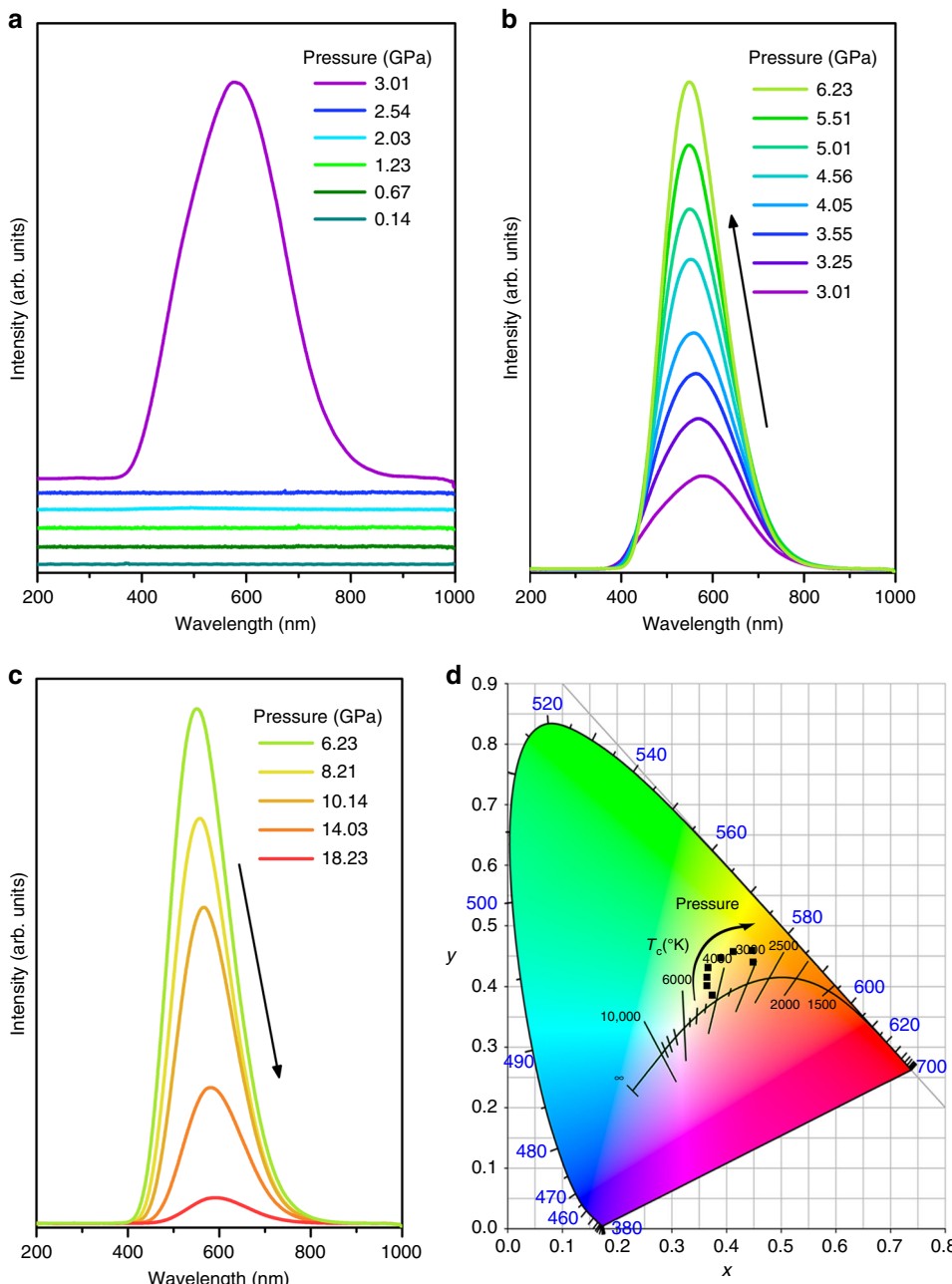

**Fig. 2** High-pressure photoluminescence properties of Cs$_4$PbBr$_6$ NCs. **a**, **b**, and **c** Changes in the PL spectra of Cs$_4$PbBr$_6$ NCs under pressure. Black arrows indicate the evolution of the PL spectra as a function of pressure. **d** Pressure-dependent chromaticity coordinates of the emissions

During this process the octahedra do not distort (Supplementary Table 1). Thus, the STE is not stabilized and no PL response is observed below 3.01 GPa (see the below analyses).

When the pressure increases beyond approximately 6.23 GPa, the PL intensity of the compressed samples decreases with increasing pressure and almost disappears at 18.23 GPa (Fig. 2c). The PL weakening should be ascribed to the deviatoric stress arising from the nonhydrostatic conditions in the diamond anvil cell (DAC), which was indeed reported in earlier studies using synchrotron radiation small-angle X-ray scattering[31–33]. Deviatoric stresses eventually lead to slow amorphization of Cs$_4$PbBr$_6$ NCs upon further compression, which is largely related to the higher degree of distortion and random orientations of inorganic octahedra within the material[25,34]. The recovered sample was also characterized by TEM, as shown in Supplementary Fig. 11. We can see that the quenched samples exhibit aggregation to a large

extent after the high-pressure treatment. It appears, therefore, that deviatoric stress should be a crucial factor in reducing the PL intensity. We further investigated the PL responses to pressure through a control high-pressure experiment by directly dispersing the samples in toluene and then loading them into the DAC (Supplementary Fig. 12)[11,13]. The samples dispersed in toluene still exhibited a fluorescence enhancement, followed by a persistent decrease in PL intensity. Note that the enhanced emission process could be maintained up to pressures of 7.89 GPa, much higher than that in the experiment adopting silicon oil as the pressure transmitting medium (PTM). Deviatoric stress also appears to exist even when the sample was loaded into the DAC in an aggregate form. The optical images (Supplementary Fig. 13) clearly demonstrate the trend of PL brightness in Cs$_4$PbBr$_6$ NCs with pressure, accompanied by color changes from bright white to dark yellow. Furthermore, we recorded the

chromaticity coordinates of emission upon compression from 3.01 to 18.23 GPa (Fig. 2d and Supplementary Table 2). Based on the CIE chromaticity diagram, the emission at 3.01 GPa lies to the yellow side of pure white-light (0.32, 0.32) with a corresponding color temperature (CCT) in the range of 3000–4500 K, resulting in warm white-light for many indoor lighting applications. In addition, the luminescence colors of the resulting $Cs_4PbBr_6$ NCs change from white to dark yellow as the pressure increases from 3.01 GPa to 18.23 GPa. Therefore, we indeed developed high-pressure technology as a robust tool to achieve not only PIE but also tuning of the chromaticity of emission. The color tunability of 0D $Cs_4PbBr_6$ NCs allows for the development of optically pumped WLEDs with different photometric properties for various applications, such as navigation lights for airplanes and military signs, where lighting specifications need to be adjusted on demand.

To elucidate the origins of the observed PIE phenomenon in $Cs_4PbBr_6$ NCs, the pressure-dependent absorption of $Cs_4PbBr_6$ NCs was determined (Fig. 3a). The absorption peak exhibits a slight redshift with increasing pressure. When the pressure increases beyond 3.02 GPa, the profile of the absorption peak undergoes a stark change at the same time as the sudden appearance of broad band emission. This distinct change in the absorption spectrum is consistent with the emission that emerged at 3.02 GPa, which is associated with the above identified pressure-induced structural phase transformation. Moreover, an additional shoulder at the low-energy side of the original absorption band appears at 3.53 GPa (Fig. 3a). As the pressure increases to 6.17 GPa, this shoulder develops into a clearly resolved sharp peak with an increased magnitude and remains stable up to 18.14 GPa. We attributed the formation of the two

absorption peaks to the splitting of the excitonic peak. As the absorption profile of perovskites is greatly influenced by the nature of the octahedra. Therefore, the formation of these two absorption peaks should be related to the distortion of $[PbBr_6]^{4-}$ octahedra. As shown in Supplementary Table 1, although the bond lengths of Pb–Br decrease with compression, the $[PbBr_6]^{4-}$ octahedra exhibit no distorted configuration below 3.01 GPa. However, in the high-pressure monoclinic phase, the three Pb–Br bonds happen to be nonequivalent during the $[PbBr_6]^{4-}$ octahedra persistent change upon compression, forming two classes of Pb–Br bonds. Therefore, it is reasonable to speculate that the breaking of the symmetry of the $[PbBr_6]^{4-}$ octahedral unit under compression leads to the splitting of the excitonic peak into two: one at higher and one at lower energies. This assignment is further supported by absorption oscillator strengths calculated by the first-principles method using the excited-state structure associated with the STE (Supplementary Fig. 14) at the single-particle level. We found that in the lower energy region the high-pressure phase shows two separated large oscillator strengths, implying the signature of absorption peak splitting, while the ambient-pressure phase shows one group of oscillator strengths. Likewise, the energy difference $\Delta E$ between the two exciton absorption peaks (Fig. 3b) can indicate the extent of distortion for octahedra with increasing pressure. As illustrated in Fig. 3c, we find that this energy difference sharply increases in the pressure region from 4.03 to 6.17 GPa with a pressure coefficient of 0.08 eV/GPa (Fig. 3c). When the pressure increases beyond 6.17 GPa, $\Delta E$ experiences a relatively slow increase. Figure 3c demonstrates that the distortion extent of octahedra is improved with compression, but beyond 6.17 GPa, the $[PbBr_6]^{4-}$ octahedra start to tilt and undergo a considerable rotation, deviating from

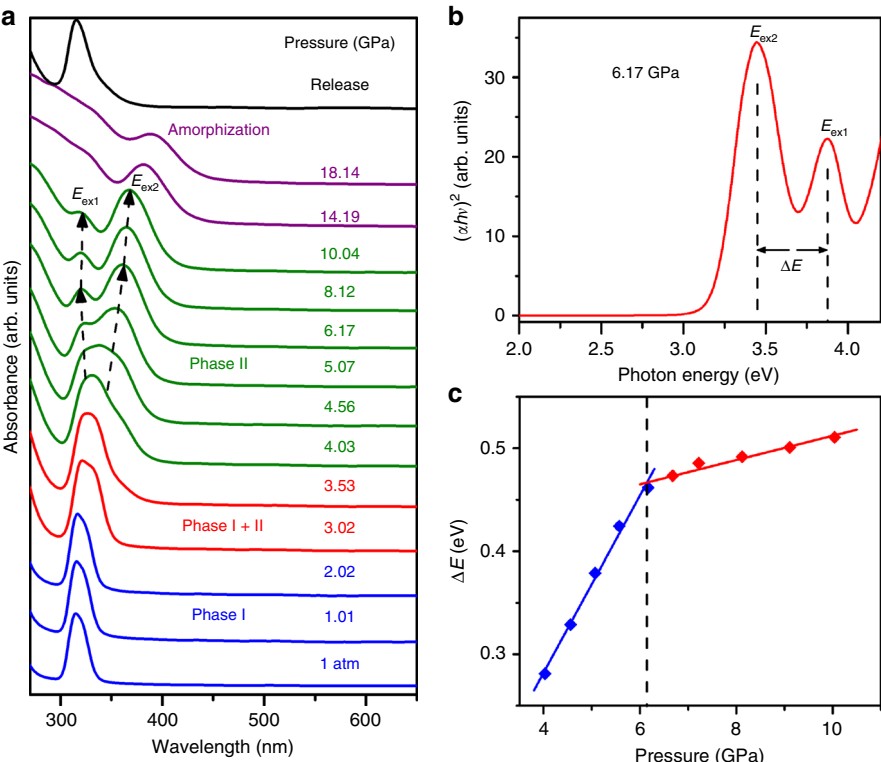

**Fig. 3** Optical absorption of $Cs_4PbBr_6$ NCs with increasing pressure. **a** Typical profile of the absorption band for $Cs_4PbBr_6$ NCs at different pressures measured in situ in a DAC apparatus. The dashed arrows indicate the shift of the bound exciton absorption peaks of $E_{ex1}$ and $E_{ex2}$. **b** Absorption spectra of $Cs_4PbBr_6$ NCs measured at a selected pressure of 6.17 GPa. $E_{ex1}$ and $E_{ex2}$ represent the splitting bound exciton absorption peaks. **c** Pressure dependence of energy difference $\Delta E$ deduced from the difference of $E_{ex1}$ and $E_{ex2}$

the original orientation in a disordered manner, consistent with the high-pressure ADXRD result (see Supplementary Fig. 8 for more details).

Moreover, because high pressure causes considerable distortion of the $[PbBr_6]^{4-}$ octahedron, the excited-state structural reorganization required to trap a photoexcited exciton is expected to be smaller than that for the ambient-pressure phase containing perfect $[PbBr_6]^{4-}$ octahedra. The resulting smaller Stokes shift may allow reasonable wavefunction overlap between the excited-state (associated with the STE) and the ground state. This may lead to a nonzero or enhanced transition dipole moment compared to the ambient-pressure phase. Given this, we calculated the transition dipole moments based on the excited-state structure associated with the STE at the single-particle level, as shown in Supplementary Figs. 1 and 4. We found that the high-pressure phase exhibits a one-magnitude larger oscillator strength (~0.06 a.u.) at the lowest excitation energy than that of the ambient-pressure phase (~0.003 a.u.). This result implies that the pressure-induced distortion of the $[PbBr_6]^{4-}$ octahedra promotes the self-trapped excitonic state to be more optically active.

In addition to some extent of enhanced optical activity, our first-principles calculations show that the STE in the high-pressure phase has the larger exciton binding energy (1.31 eV) than that in the ambient-pressure phase (1.13 eV). This phenomenon originates from the increased electron-phonon coupling strength when the $[PbBr_6]^{4-}$ octahedra are contracted [accompanied by the strengthened Pb–Br covalent bondings (Supplementary Table 1)] upon compression. Because the STE is mediated by the interaction between the exciton and lattice distortion, the stronger electron-phonon coupling may more effectively bind photoexcited carriers to form STEs. The Supplementary Fig. 15 shows the comparison of the electronic band structures with and without the lattice distortion mediating STEs for the ambient-pressure (1 atm) and high-pressure (4 GPa). As seen, the electronic band structure is more seriously distorted in the presence of lattice distortion for the high-pressure phase, which ambiguously indicates a stronger electron-phonon coupling. Such increased electron-phonon coupling strength is responsible for the larger binding energy of the STE in the high-pressure phase.

Figure 4 depicts a schematic of the emission processes in $Cs_4PbBr_6$ NCs with and without compression. The excitation transition from A to B is described. Due to the unique structure of $Cs_4PbBr_6$ NCs, the excited carriers are readily localized to form bound excitons from the conduction band due to strong quantum confinement. The formed bound excitons subsequently relax to the self-trapped state via B to C. Under ambient conditions, the ideal octahedral structures have low activation energy for detrapping due to the weak electron-phonon coupling strength. Therefore, the photoexcited carriers are readily detrapped from the self-trapped state to the bound exciton state by thermal activation. In addition, the low optical activity mentioned above also hinders the appearance of emission. However, at 4 GPa, the distorted octahedra within high-pressure monoclinic $Cs_4PbBr_6$ NCs result in a decrease in the Stokes shift, which allows reasonable wavefunction overlap between the excited-state (associated with the STE) and the ground state. This leads to enhanced optical activity compared to the ambient-pressure phase. In addition, the distorted octahedral structure possesses a stronger electron-phonon coupling strength (the Huang-Rhys parameter $S^{1/2}$ can reflect the strength of electron-phonon coupling because $S^{1/2}$ generally increases with an increase in the electron-phonon coupling strength), resulting in an enhancement of the activation energy for detrapping, and thus, the STEs can hardly convert to bound excitons via thermal activation. Subsequently, the broad emission associated with the transition of the STEs to the valence band through radiative recombination, as illustrated from C to D (Fig. 4b), appears. With much higher pressure, the $[PbBr_6]^{4-}$ octahedra are further contracted, and the Pb–Br covalent bondings are additionally strengthened (Supplementary Table 1), which further enhances the electron-phonon coupling strength. This enables an increase in the concentration of STEs, thus improving the possibility of radiative recombination. Therefore, a persistent increase in PL intensity can be observed with compression.

In summary, we have investigated the structural evolution and optical response of NCs of the 0D all-inorganic MHP $Cs_4PbBr_6$ as a function of pressure up to 18 GPa. Intriguingly, PIE and a subsequent large emission enhancement behavior were achieved upon compression. The in situ ADXRD data and Rietveld refinement results indicate that the unexpected emission is

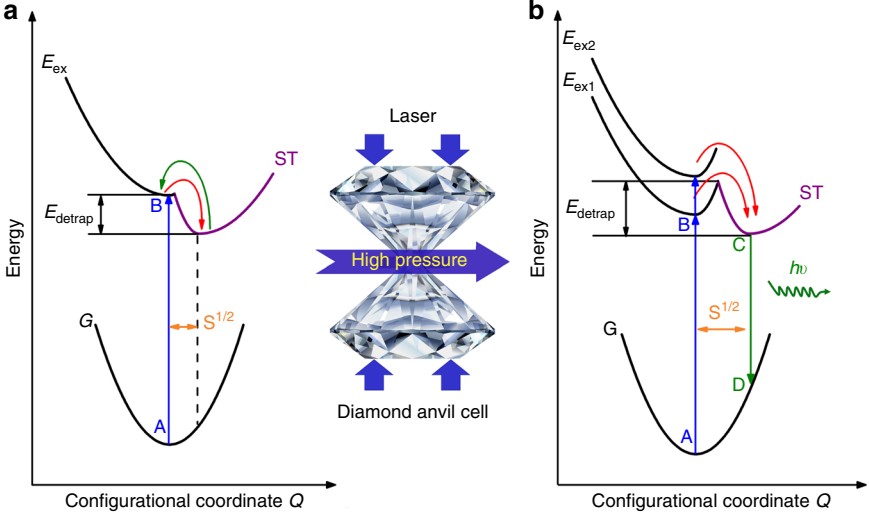

**Fig. 4** Pressure-induced emission mechanism associated exciton self-trapping in NCs. Configuration coordinate model of emission for the $Cs_4PbBr_6$ NCs at 1 atm (**a**) and 4 GPa (**b**). Herein, the absorption transition upon excitation from A to B is described. The STE recombination emission is depicted from C to D. The path between B and C refers to exciton self-trapping (red) and detrapping (green). ($E_{detrap}$): activation energy for detrapping, ($E_{ex1}$ and $E_{ex2}$): splitting of the bound exciton state, (ST): self-trapped state, (G): ground state, $S^{1/2}$: Huang-Rhys parameter

associated with the deformation of $[PbBr_6]^{4-}$ octahedra in the newly emerged high-pressure monoclinic phase, which is also supported by first-principles energetic calculations. We ascribe the PIE to radiative recombination of the self-trapped excitons associated with the large distortion of the Pb–Br octahedra after a phase transition. Joint experimental and theoretical analyses indicate that the emergence of room-temperature luminescence may be ascribed to the enhanced optical activity and the increased binding energy of self-trapped excitonic states under high pressure. Our findings not only provide the fundamental relationship between structural variations and the optoelectronic properties of $Cs_4PbBr_6$ NCs but also offer insight into the microscopic physiochemical mechanisms of MHP nanosystems at extremes.

## Methods

**Fabrication of $Cs_4PbBr_6$ nanocrystals**. $Cs_4PbBr_6$ NCs were synthesized according to a modified synthetic approach[8]. In a typical synthesis, cesium carbonate (0.4 g, $Cs_2CO_3$, Sigma-Aldrich, 99.9%) and oleic acid (8 mL, OA, Sigma-Aldrich, 90%) were loaded into a 50 mL three-neck flask. After degassing for 20 min, the solvent was stirred at 130 °C under $N_2$ for 1 h to generate a yellowish Cs-oleate complex, and then was heated under $N_2$ to 150 °C. Meanwhile, the mixture of $PbBr_2$ (0.1468 g, Sigma-Aldrich, 99%), 20 mL of octadecene (ODE, technical grade, 90%), 6 mL of oleylamine (OLAM, 70%) and 0.8 mL of OA were loaded into a 50 mL three-neck flask. After degassing for 20 min, the solution was vigorously stirred on a hotplate set at 150 °C. After complete solubilization of $PbBr_2$, the temperature was reduced to 80 °C and 3 mL of Cs-oleate solution was swiftly injected. The turbid white reaction mixture was extracted after 3 min and quenched by toluene. The resultant mixture was isolated using toluene by centrifuging for 10 min at 4500 rpm.

**Characterization and high-pressure generation**. We characterized our samples by transmission electron microscopy (TEM) and high-resolution TEM performed on a JEM-2200FS with an emission gun operating at 200 kV. High-pressure experiments were carried out with a symmetric DAC. The sample and a small ruby ball were loaded into the 150 μm-diameter chamber of a DAC, constructed from a T301 steel gasket pre-indented to a thickness of 45 μm. The pressure calibration was determined utilizing the standard ruby fluorescent technique[35]. In high-pressure experiments, silicon oil was utilized as the pressure transmitting medium (PTM) for optical absorption, PL and XRD experiments, while the argon was employed as PTM for Raman measurements. These PTM did not have any detectable effect on the behavior of $Cs_4PbBr_6$ NCs under pressure. All of the measurements were performed at room temperature.

**In situ high-pressure experiments**. The excitation source, a 355 nm line of a UV DPSS laser with the power of 4.5 mW, was used for PL measurements. The high-pressure evolution of steady-state PL spectra of $Cs_4PbBr_6$ NCs was collected by a modified spectrophotometer (Ocean Optics, QE65000) with the data-collection time of 5 s. The laser beam passed through tunable filter and was focused onto the sample with 20 μm spot 20 × UV Plan apochromatic objective. Each new acquisition was carried out several minutes later after elevation of the pressure, aiming to account for any kinetic dependence during measurements. PL micrographs of the samples were obtained using a camera (Canon Eos 5D mark II) equipped on a microscope (Ecilipse TI-U, Nikon). The camera can record the photographs under the same conditions including exposure time and intensity. The chromaticity coordinates (x, y) were calculated from the fluorescence data (400–700 nm) using the CIE1931xy.V.1.6.0.2a software package. The color of the fluorescence emissions were identified by the CIE colorimetry system. Any color could be described by the chromaticity (x, y) coordinates on the CIE diagram. The in situ high-pressure absorption spectra were measured by a deuterium-halogen light source and recorded with an optical fiber spectrometer (Ocean Optics, QE65000). In situ high-pressure angle-dispersive X-ray diffraction (ADXRD) patterns were obtained with a wavelength of 0.6199 Å at beamline 15U1, Shanghai Synchrotron Radiation Facility (SSRF), China. $CeO_2$ was used as the standard sample to do the calibration. The collected 2D images were integrated on the basis of FIT2D program, yielding 1D intensity versus diffraction angle 2-theta pattern. All the high-pressure experiments were conducted at room temperature. In situ high-pressure Raman spectra were recorded using a spectrometer equipped with liquid nitrogen cooled CCD (iHR 550, Symphony II, Horiba Jobin Yvon). A 785 nm single-mode DPSS laser was utilized to excite the sample, and the output power was 10 mW. The resolution of the system was 1 cm$^{-1}$.

**First-principles calculations**. Calculations were performed within the framework of density functional theory (DFT) by using plane-wave pseudopotential methods as implemented in the Vienna Ab initio Simulation Package. The electron–ion interactions were described by the projected augmented-wave pseudopotentials[36] with the 6s (Cs), 6s and 6p (Pb), 4s and 4p (Br) electrons treated explicitly as valence electrons. We used the generalized gradient approximation formulated by Perdew, Burke, and Ernzerhof as the exchange correlation functional. Kinetic energy cutoff for the plane-wave basis set was set to 520 eV. The k-point meshes with grid spacing of $2\pi \times 0.15/\text{Å}$ or less were used for electronic Brillouin zone integration. The equilibrium structures at different pressures were optimized through total energy minimization with the residual forces on the atoms converged to below 0.01 eV/Å.

For electronic band structure and STE calculations, we used hybrid functional of Hyed-Scuseria-Ernzerhof (HSE) approach[37] to remedy the self-interaction error of DFT calculations. The spin-orbit coupling (SOC) effect was included. In the STE calculations, we constructed the 88-atom supercell that can reliably describe stable polaronic states of both electron and hole carriers. This involves a local symmetry-breaking octahedron distortion (i.e. extending two Pb–Br bonds around a single $[PbBr_6]$ octahedron and decreasing other four Pb–Br bonds on the equatorial plane). The polaronic states were simulated with a more accurate residual force threshold of 0.005 eV/Å. We calculated the electronic configuration of the STE structure with a charge-neutral supercell where an electron is constrained to occupy the conduction band, leaving a hole at the valence band. The lowest-energy spin-triplet state (with the same spin directions of electron and hole) is adopted. The binding energy of STE (relative to a free exciton) was calculated by the equation: $E_b = E(GS) + E_g - E(\text{exciton})$, where $E(GS)$ and $E(\text{exciton})$ are the total energies of the ground state and the STE state, respectively, and $E_g$ is the band gap. Our results on the ambient-pressure phase of $Cs_4PbBr_6$ are qualitatively in agreement with the previous studies[21,38], though different exchange correlation functionals were used.

## Data availability

The data that support the findings of this study are available from the corresponding authors on request.

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

## Acknowledgements

This work is supported by National Key R&D Program of China (No. 2018YFA0305900), the National Science Foundation of China (Nos. 21725304, 11774125, 21673100, 91227202, and 11774120), the Chang Jiang Scholars Program of China (No. T2016051), Changbai Mountain Scholars Program (No. 2013007), National Defense Science and Technology Key Laboratory Fund (6142A0306010917), Scientific Research Planning Project of the Education Department of Jilin Province (JJKH20180118KJ), and Program for JLU Science and Technology Innovative Research Team. S.A.T.R. acknowledges the support of the NERC grant NE/P019714/1. X.F. is grateful for the support of the China Scholarship Council. L.Z. acknowledges the support of the National Science Foundation of China (Grant 61722403 and 11674121), National Key Research and Development Program of China (Grant 2016YFB0201204). Calculations were performed in part at High Performance Computing Center of Jilin University. This work was performed at beamline 15U1, Shanghai Synchrotron Radiation Facility (SSRF), and 4W2 HP-Station, Beijing Synchrotron Radiation Facility (BSRF), which is supported by Chinese Academy of Science (grant nos. KJCX2-SW-N20 and KJCX2-SW-N03).

## Author contributions

G.X., L.Z., and B.Z. proposed the research direction and guided the project. Z.M., G.X., and B.Z. carried out the experiments and analyzed the experimental data. Z.M., Z.L., S.L., D.Y., and L.Z. performed the first-principles calculation and analyzed the theoretical data. L.W. and K.W. assisted in performing partial experiments. Z.M., X.F., G.X., L.Z., S.A.T.R., and B.Z. discussed the results and participated in writing the manuscript.

## Additional information

**Competing interests:** The authors declare no competing interests.

