## [Peer Review File · Nature Communications]

Reviewers' Comments:

Reviewer #1:

Remarks to the Author:

Pressure processing is emerging as one significant tool to make quick discovery of advanced materials with novel structures and transformative properties. In the case of newly emergent metal halide perovskites with a record-high conversion efficiency, the authors combined pressure processing approach with in-situ spectroscopic techniques, and observed the emergence of strong emission as Cs₄PbBr₆ nanocrystals (NCs) was compressed to 3.01 GPa. Surprisingly, continuous compression caused a dramatic enhancement of PL intensity, which reached maximum at 6.23 GPa, and then weakened gradually. In combination with synchrotron x-rays, Raman spectroscopy and theoretical computations, the structural transformations were used to provide a reasonable interpretation. As a completely new phenomenon which has potential application, this work certainly deserves a publication at Nature Communication.

Below are several concerns and/or suggestions for revision:

1. For the datasets presented in Figure 2d and Table S1, the authors should provide a detailed description of how the chromaticity coordinates (x, y) were calculated from their PL measurements. The authors should also describe a bit about what kind of typical information that Figure 2d conveys to the audiences for better understanding the essence, significance and application of presented datasets.
2. The authors need to provide much more details of experimental and analytical processes about the PL measurement, for example, the laser focus spot, the exposure and data-collection time, the peak intensity and FWHM, and so on. These details are extremely important and helpful for other researchers to reproduce and quantify the data sets as presented in this work.
3. As shown in Figure S10, the loaded samples are clearly NC aggregates, rather than completely dispersed NCs in solvents. Such a sample loading way inevitably results in the generation of deviatoric stress across the samples under higher pressures. If look at recent high pressure SAXS and WAXS works on NCs (Wang T. et al., *Adv. Mater.*, 2015, 27, 4544; Zhu, J. et al, *Nanoscale*, 2016, 8, 5214; Wang Z. et al., *JACS*, 2011, 133, 14484), much higher sensitivity of SAXS technique allows one to easily see the pressure-induced appearance of deviatoric stress from the compressed samples (indeed, not very high). Even though the samples are surrounded by silicone oil, through the pressure-dependent plot of SAXS datasets, we can easily find the appearance of deviatoric stress from the samples. Therefore, the observed weakening of PL emission at pressures above 6.8 GPa is most likely related to a nonhydrostatic effect. The authors should combine previous SAXS works to discuss such an effect.
4. Based on the above argument on deviatoric stress, it is most likely that another nano-based form formed in such Cs₄PbBr₆ perovskite NCs under high pressure (see the references: Nagaoka, Y et al., *Adv. Mater.*, 2017, 29, 1606666; Wu, H. et al., *Angewandte*, 2010, 49, 8431). I suggest that the authors check the recovered sample by TEM. Alternatively, the authors could also add one control experiment, in which NCs are dispersed in a good solvent (such as toluene) and loaded into DAC to check how PL responses with pressure.

Reviewer #2:

Remarks to the Author:

The manuscript deals with pressure dependence of the structural and luminescence properties of Cs₄PbBr₆. There is novelty in this work and the results are quite interesting. In my opinion, the findings may be useful in obtaining an understanding of the intriguing luminescence properties of Cs₄PbBr₆. While I am quite positive about the manuscript and like to see it published in this

journal, the authors need to address the following points and revise the manuscript appropriately before it can be recommended for publication.

1. Why does the exciton binding energy increase with increase in pressure?
2. Why the VB and conduction band edge energy levels or self-trapped state energy level not affected by pressure?
3. As the self-trapped state is of specific energy, the broad nature of the emission is not understandable.
4. Insight into the formation 'self-trapped state' due to pressure-induced distortion of the octahedron is lacking.
5. Why do the excited carriers bind readily to the free exciton state when the nanocrystals are in their monoclinic phase (sentence 218-220)?

Minor points:

There are a number of grammatical and typographical errors which are to be corrected.

Reviewer #3:

Remarks to the Author:

This paper reports on the pressure induced emission of halide perovskite Cs₄PbBr₆ nanocrystals (OD NCs). The authors show that films of non-emitting OD NCs can emit once they are exposed to high pressure up to 3.01 GPa. The authors attribute such emission to self-trapped electrons from highly distorted [PbBr₆]⁴⁻ octahedra at high pressures.

This paper is surely interesting and might be of potential interest for a wide audience. On the other hand, considering the ongoing debate about the origin of the emission for these systems, the authors provide very little evidence about what produces such emission. In addition, the explanation of the emission mechanism is rather confusing and not clear. In short, the authors do not provide sufficient and compelling explanation on the origin of the emission of these systems at high pressure. To make this work publishable, the authors should be less vague on their claims and be able to prove what they speculate about.

Below I explain my major concerns with the manuscript:

1. From line 90-95, the authors attribute the mid-gap state emission in OD NC (at 1 atm) to state derived from halogen vacancy. This is actually not true. The halogen vacancy is one of the many theories that explain emission, and very likely the least accurate, considering the high presence of halogen in this system (more than 6:1 vs Pb). So far, the more robust explanation for the emission is attributed to the presence of 3D perovskite impurities embedded inside the Cs₄PbBr₆ matrix. See, for example, Manna's papers for details. This explains also why emitting OD powders absorbs at around 500nm at energies similar to the emission.
2. In lines 52-55, the authors say that the emission is related to the Jahn-Teller emission due to distorted octahedral. However, later in the manuscript and in the supporting info, the authors show, using DFT calculations, that the bandgap is reduced from 3.23 eV to 3.14 eV, only 0.09 eV, therefore indicating that despite the distortion, the change in energy is very minor.
3. In the caption of Figure 2, the authors mention absorption and PL emission, however the graphs seem to be only related to emission.
4. From line 125-129, the authors analyze the Raman spectra to explain the behavior of low vibrational mode. While this is per se a good idea, the whole explanation is quite confusing and not sufficiently clear. There is lots of wording, but not clear insights. Especially the sentence "local

fluctuation ... will induce the formation of small polarons, resulting in trapping of excitons by the lattice". What does it mean? Are you actually proving that these polarons are formed and they have consequences on trapping? This seems to me just speculation and not a concrete analysis.

5. In lines 132-134, the authors say that the electron-phonon coupling give rise to a highly distorted excited state. This sentence however is not backed-up by neither the experiments nor theory (no calculations have been done to look at this effect). On the other hand, a large Stokes shift might be a better and more reasonable explanation for the emission. The Jahn-Teller effect that would occur in the excited state might be indeed quite large. However, the fact that OD NC are non-emissive works against this explanation. And DFT calculations show that the Jahn-Teller distortion is around ~ 1 eV, not enough to exhibit emission in the ~ 520 nm region. Additionally, the authors should prove that at high pressure such emission becomes optically active in terms of transition dipole moments (demonstrated by DFT calculations for example). In short, why without pressure, there is no emission, while with pressure there is emission if the Jahn-Teller is responsible to the emission ?

6. The mechanism of PIE under high pressure provided from line 180 is very much unclear and based on fuzzy terminology. The authors attribute the appearance of two peaks at high pressure as the exciton energy and energy gap (?!?). What does this mean ??? Why invoking these terms ? In bulk structures, you have a band gap, and in molecular like structures you have the optical gap. In NC, you have more than the latter than the former. Especially, for such OD NC system, which will always exhibit a molecular-like absorption, even in the bulk, because of the disconnection between the octahedra. Meaning the exciton is always confined inside the octahedral (self-trapped). The authors should be more clear in the definition of the terms they introduce. Seeing two peaks and arbitrarily attributing them to exciton and band gap transitions, whatever it means, and even extrapolating the exciton binding energy is quite irritating. As such, also Figure 4 does not make much sense.

7. The appearance of two peaks can be attributed to for example, two types of phases with different absorption onsets. The XRD spectra in the supporting info also indicates that this phase is quite amorphous, meaning that broadening and splitting of peaks are very likely. Another explanation to the emission is that under pressure, you may distort so much the lattice that you could lead to connection between octahedra, and the formation of impurities of 3D structures. You cannot discard this for two reasons: the red shift in absorption and the amorphous structure you find at these high pressures.

Dear Reviewer: 1

First of all, we acknowledge your comments and suggestions very much, which are valuable in improving the quality of our manuscript. We revised our manuscript in accordance with your instructive guidance and we feel that the revised manuscript is a great improvement on the original.

Comments: Pressure processing is emerging as one significant tool to make quick discovery of advanced materials with novel structures and transformative properties. In the case of newly emergent metal halide perovskites with a record-high conversion efficiency, the authors combined pressure processing approach with in-situ spectroscopic techniques, and observed the emergence of strong emission as Cs₄PbBr₆ nanocrystals (NCs) was compressed to 3.01 GPa. Surprisingly, continuous compression caused a dramatic enhancement of PL intensity, which reached maximum at 6.23 GPa, and then weakened gradually. In combination with synchrotron x-rays, Raman spectroscopy and theoretical computations, the structural transformations were used to provide a reasonable interpretation. As a completely new phenomenon which has potential application, this work certainly deserves a publication at Nature Communication.

Comment 1: For the datasets presented in Figure 2d and Table S1, the authors should provide a detailed description of how the chromaticity coordinates (x, y) were calculated from their PL measurements. The authors should also describe a bit about what kind of typical information that Figure 2d conveys to the audiences for better understanding the essence, significance and application of presented datasets.

Author reply: We are grateful for the reviewer's expertise and kind suggestions. According to the advice, we have added the detailed description of how the chromaticity coordinates (x, y) were calculated from PL measurement, and introduced more information about the presented datasets. The Commission Internationale de L'Eclairage coordinate is the most common method to describe the composition of any color in terms of three primaries. The so-called chromaticity coordinates x, y, z are denoted by the ratios of tristimulus values X, Y, Z of the light to their sum. Therefore, it only needs to quote the two quantities of the reference stimuli to define a color since the three quantities (x, y, z) are summed to 1.¹ The description of the PL in these terms would give an objective measure of the visual perception of the colors, and enable quantitative specifications of the magnitude of perceived color differences or changes. Herein, chromaticity coordinates (x, y) in Figure 2d were calculated from the corresponding fluorescence spectra using the CIE1931xy.V.1.6.0.2a software package, which changed from (0.3734, 0.3852) at 3.01 GPa to (0.4500, 0.4397) at 18.23 GPa (see Table S2). Based on the CIE chromaticity diagram, the emission at 3.01 GPa lies

to the yellow side of pure white light (0.32, 0.32) with a corresponding color temperature (CCT) in the range of 3000-4500 K, resulting in the “warm” white light for many indoor lighting applications. In addition, the luminescence colors of the resulting Cs₄PbBr₆ NCs changes from white to dark yellow as the pressure increases from 3.01 GPa to 18.23 GPa. Therefore, we indeed developed high-pressure technology as a robust tool to achieve not only PIE, but also to tune chromaticity of emission. The color tunability of 0D Cs₄PbBr₆ NCs allows for the development of optically-pumped WLEDs with different photometric properties for various applications, such as navigation lights for airplanes and military signs, where lighting specifications need to be adjusted on demand.

The revised details are shown as follows:

Experimental Section:

The chromaticity coordinates (x, y) were calculated from the fluorescence data (400-700 nm) using the CIE1931xy.V.1.6.0.2a software package. The color of the fluorescence emissions were identified by the CIE colorimetry system. Any color could be described by the chromaticity (x, y) coordinates on the CIE diagram.

The corresponding revised details are highlighted in red and can be found in **Line 19-23, Page 15** of the revised Experimental Section.

2.2 In situ high-pressure photoluminescence measurement of Cs₄PbBr₆ NCs

Furthermore, we recorded the chromaticity coordinates of emission upon compression from 3.01 to 18.23 GPa (Figure 2d and Table S2). Based on the CIE chromaticity diagram, the emission at 3.01 GPa lies to the yellow side of pure white light (0.32, 0.32) with a corresponding color temperature (CCT) in the range of 3000-4500 K, resulting in the “warm” white light for many indoor lighting applications. In addition, the luminescence colors of the resulting Cs₄PbBr₆ NCs changes from white to dark yellow as the pressure increases from 3.01 GPa to 18.23 GPa. Therefore, we indeed developed high-pressure technology as a robust tool to achieve not only PIE, but also to tune the chromaticity of emission. The color tunability of 0D Cs₄PbBr₆ NCs allows for the development of optically-pumped WLEDs with different photometric properties for various applications, such as navigation lights for airplanes and military signs, where lighting specifications need to be adjusted on demand.

The corresponding revised details are highlighted in red and can be found in **Line 18-30, Page 9** of the revised manuscript.

REFERENCE

(1) Ye, S., Xiao, F., Pan, Y. X., Ma, Y. Y. & Zhang, Q. Y. Phosphors in phosphor-converted white light-emitting diodes: Recent advances in materials, techniques and properties. *Mat. Sci. Eng. R* 71, 1-34, (2010).

Comment 2: *The authors need to provide much more details of experimental and analytical processes about the PL measurement, for example, the laser focus spot, the exposure and data-collection time, the peak intensity and FWHM, and so on. These details are extremely important and helpful for other researchers to reproduce and quantify the data sets as presented in this work.*

Author reply: Thanks very much for the reviewer's insightful comments. According to the reviewer's advice, we have revised the manuscript and introduced more descriptions about experimental process about the PL measurement in experimental section.

The revised details are shown as follows:

Experimental Section:

The excitation source, a 355 nm line of a UV DPSS laser with the power of 4.5 mW, was used for PL measurements. The high-pressure evolution of steady-state PL spectra of Cs₄PbBr₆ NCs was collected by a modified spectrophotometer (Ocean Optics, QE65000) with the data-collection time of 5 s. The laser beam passed through tunable filter and was focused onto the sample with 20 μm spot 20× UV Plan apochromatic objective. Each new acquisition was carried out several minutes later after elevation of the pressure, aiming to account for any kinetic factor during the measurements. PL micrographs of the samples were obtained using a camera (Canon Eos 5D mark II) equipped on a microscope (Ecclipse TI-U, Nikon). The camera can record the photographs under the same conditions including exposure time and intensity.

The corresponding revised details are highlighted in red and can be found in **Line 9-19, Page 15** of the revised Experimental Section.

2.2 In situ high-pressure photoluminescence measurement of Cs₄PbBr₆ NCs

Note that the corresponding PL intensity shows a remarkable increase with increasing pressure, eventually reaching a maximum at the pressure of 6.23 GPa (Figure 2b, Figure S3a). Pressure dependence of the PL wavelength and the full width half maximum (FWHM) in our Cs₄PbBr₆ NCs is shown in Figure S3b and S3c. We can see that the PL peak of Cs₄PbBr₆ NCs shows an initial blue-shift upon compression to 6.23 GPa. Thereafter, it displays a redshift to 18.23 GPa, at which point the PL intensity almost disappeared

(Figure 2c). The FWHM of the PL peak sharply decreases in the pressure region from 3.01 to 6.17 GPa, followed by a relatively sluggish increase.

The corresponding revised details are highlighted in red and can be found in Line 7-15, Page 7 in the revised manuscript, as well as Figure S3 in Supporting Information.

Added Figure S3. (a) Normalized PL intensity as a function of pressure. (b) The PL location of Cs₄PbBr₆ NCs against pressures. (c) Full width at half-maximum (FWHM) vs pressure.

Comment 3: As shown in Figure S10, the loaded samples are clearly NC aggregates, rather than completely dispersed NCs in solvents. Such a sample loading way inevitably results in the generation of deviatoric stress across the samples under higher pressures. If look at recent high pressure SAXS and WAXS works on NCs (Wang T. et al., *Adv. Mater.*, 2015, 27, 4544; Zhu, J. et al, *Nanoscale*, 2016, 8, 5214; Wang Z. et al., *JACS*, 2011, 133, 14484), much higher sensitivity of SAXS technique allows one to easily see the pressure-induced appearance of deviatoric stress from the compressed samples (indeed, not very high). Even though the samples are surrounded by silicone oil, through the pressure-dependent plot of SAXS

datasets, we can easily find the appearance of deviatoric stress from the samples. Therefore, the observed weakening of PL emission at pressures above 6.8 GPa is most likely related to a nonhydrostatic effect. The authors should combine previous SAXS works to discuss such an effect.

Author reply: We are grateful for the reviewer's expertise and kind suggestions. We carefully read the literatures that the reviewer provided, and conclude that the deviatoric stress should be largely related to the observed weakening of PL emission at pressures above 6.8 GPa. We also make some revisions and add further discussions about the nonhydrostatic effect in the manuscript by combining the previous SAXS works.

The revised details are stated as follows:

When the pressure was beyond roughly 6.23 GPa, the PL intensity of the compressed samples decreased with pressure, and almost disappears at 18.23 GPa (Figure 2c). The weakening of PL should be ascribed to the deviatoric stress arising from nonhydrostatic conditions in the diamond anvil cell (DAC), which was indeed reported from earlier studies using small-angle x-ray scattering (SAXS).³¹⁻³³ Deviatoric stresses eventually lead to sluggish amorphization of Cs₄PbBr₆ NCs upon further compression, which was largely related to the higher degree of distortion and random orientation in inorganic octahedra.^{24,34}

The nonhydrostatic effect is an important factor for pressure-induced amorphization in perovskite materials.⁶ The loaded samples are NC aggregates, which possibly results in the generation of deviatoric stress across the Cs₄PbBr₆ NCs under higher pressures. Therefore, the nonhydrostatic effect cannot be excluded to interpret pressure-induced the weakening of PL. Above 6.8 GPa, all the diffraction peaks start to broaden and merge, likely due to the deviatoric stress induced amorphization. Additionally, the TEM image of the decompressed sample (Figure S13) indicates a large aggregation and a high extent of deformation, which is a straightforward evidence for the above-claimed conclusion. Thereby, aggregation-resulted nonhydrostatic effect should be a crucial factor for the weakening of PL intensity.

The revised details are highlighted in red and can be found in **Line 27-30, Page 8 and Line 1-5, Page 9** of the revised manuscript, and in **Line 4-14, Page 15** of the revised Supporting Information.

REFERENCE

- (24) Postorino, P. & Malavasi, L. Pressure-Induced Effects in Organic-Inorganic Hybrid Perovskites. *J. Phys. Chem. Lett.* **8**, 2613-2622, (2017).
- (31) Wang, T. *et al.* Pressure Processing of Nanocube Assemblies Toward Harvesting of a Metastable PbS Phase. *Adv. Mater.* **27**, 4544-4549, (2015).

- (32) Zhu, J. *et al.* Structural evolution and mechanical behaviour of Pt nanoparticle superlattices at high pressure. *Nanoscale* **8**, 5214-5218, (2016).
- (33) Wang, Z. *et al.* Deviatoric Stress Driven Formation of Large Single-Crystal PbS Nanosheet from Nanoparticles and in Situ Monitoring of Oriented Attachment. *J. Am. Chem. Soc.* **133**, 14484-14487, (2011).
- (34) Wang, Y. *et al.* Pressure-Induced Phase Transformation, Reversible Amorphization, and Anomalous Visible Light Response in Organolead Bromide Perovskite. *J. Am. Chem. Soc.* **137**, 11144-11149, (2015).
- (6) Postorino, P. & Malavasi, L. Pressure-Induced Effects in Organic-Inorganic Hybrid Perovskites. *J. Phys. Chem. Lett.* **8**, 2613-2622, (2017).

Added Figure S13. TEM (a) images of the originally starting Cs₄PbBr₆ NCs. TEM (b) images of the decompressed Cs₄PbBr₆ NCs when the pressure was released to the ambient conditions.

Comment 4: *Based on the above argument on deviatoric stress, it is most likely that another nano-based form formed in such Cs₄PbBr₆ perovskite NCs under high pressure (see the references: Nagaoka, Y *et al.*, *Adv. Mater.*, 2017, 29, 16066666; Wu, H. *et al.*, *Angewandte*, 2010, 49, 8431). I suggest that the authors check the recovered sample by TEM. Alternatively, the authors could also add one control experiment, in which NCs are dispersed in a good solvent (such as toluene) and loaded into DAC to check how PL responses with pressure.*

Author reply: Thanks very much for the reviewer's constructive comments. According to the advice, we checked the recovered sample by TEM, as shown in Figure S13. We can see that the quenched samples exhibit an aggregation, to a large extent, after the high-pressure treatment. It appears, therefore, that deviatoric stress should be a crucial factor in reducing the PL intensity, as the reviewer pointed out as mentioned above. We further investigated the PL responses with pressure through a control high-pressure experiment by directly dispersing the samples in toluene, and then loading it into the DAC. Figure S14 shows selected steady-state PL spectra with increasing pressure at room temperature. It was found that the

samples dispersed in toluene still exhibit a fluorescence enhancement, followed by persistent decrease in PL intensity. Note that the enhanced emission process could be maintained up to the pressure of 7.89 GPa, much higher than that in experiment by adopting silicon oil as pressure transmitting medium (PTM). Deviatoric stress also appears to exist even when the sample was loaded into the DAC in an aggregation form.

We have made some modifications in our revised manuscript. The revised details are stated as follows:

When the pressure was beyond roughly 6.23 GPa, the PL intensity of the compressed samples decreased with pressure, and almost disappears at 18.23 GPa (Figure 2c). The weakening of PL should be ascribed to the deviatoric stress arising from nonhydrostatic conditions in the diamond anvil cell (DAC), which was indeed reported from earlier studies using small-angle x-ray scattering (SAXS).³¹⁻³³ Deviatoric stresses eventually lead to sluggish amorphization of Cs₄PbBr₆ NCs upon further compression, which was largely related to the higher degree of distortion and random orientation in inorganic octahedra.^{24,34} Recovered sample was also characterized by TEM, as shown in Figure S13. We can see that the quenched samples exhibit an aggregation, to a large extent, after the high-pressure treatment. It appears, therefore, that deviatoric stress should be a crucial factor in reducing the PL intensity. We further investigated the PL responses with pressure through a control high-pressure experiment by directly dispersing the samples in toluene, and then loading it into the DAC (Figure S14).^{11,13} It was found that the samples dispersed in toluene still exhibited a fluorescence enhancement, followed by persistent decrease in PL intensity. Note that the enhanced emission process could be maintained up to pressures of 7.89 GPa, much higher than that in experiment by adopting silicon oil as pressure transmitting medium (PTM). Deviatoric stress also appears to exist even when the sample was loaded into the DAC in an aggregation form.

The nonhydrostatic effect is an important factor for pressure-induced amorphization in perovskite materials.⁶ As the loaded samples are NC aggregates, this possibly results in the generation of deviatoric stress across the Cs₄PbBr₆ NCs under higher pressures. Therefore, the nonhydrostatic effect cannot be excluded to interpret pressure-induced the weakening of PL. Above 6.8 GPa, all the diffraction peaks started to broaden and merge, likely due to the deviatoric stress induced amorphization. Additionally, the TEM image of the decompressed sample (Figure S13) indicates a large aggregation and a high extent of deformation, which is a straightforward evidence for the above-claimed conclusion. Thereby, aggregation-resulted nonhydrostatic effect should be a crucial factor for the weakening of PL intensity.

The revised details are highlighted in red and can be found in Line 27-30, Page 8 and Line 1-16, Page 9 of the revised manuscript and in Line 4-14, Page 15 of the revised Supporting Information

REFERENCE

- (11) Wu, H. *et al.* Pressure-Driven Assembly of Spherical Nanoparticles and Formation of 1D-Nanostructure Arrays. *Angew. Chem. Int. Ed.* **49**, 8431-8434, (2010).
- (13) Yasutaka, N. *et al.* Nanocube Superlattices of Cesium Lead Bromide Perovskites and Pressure-Induced Phase Transformations at Atomic and Mesoscale Levels. *Adv. Mater.* **29**, 1606666, (2017).
- (24) Postorino, P. & Malavasi, L. Pressure-Induced Effects in Organic–Inorganic Hybrid Perovskites. *J. Phys. Chem. Lett.* **8**, 2613-2622, (2017).
- (31) Wang, T. *et al.* Pressure Processing of Nanocube Assemblies Toward Harvesting of a Metastable PbS Phase. *Adv. Mater.* **27**, 4544-4549, (2015).
- (32) Zhu, J. *et al.* Structural evolution and mechanical behaviour of Pt nanoparticle superlattices at high pressure. *Nanoscale* **8**, 5214-5218, (2016).
- (33) Wang, Z. *et al.* Deviatoric Stress Driven Formation of Large Single-Crystal PbS Nanosheet from Nanoparticles and in Situ Monitoring of Oriented Attachment. *J. Am. Chem. Soc.* **133**, 14484-14487, (2011).
- (34) Wang, Y. *et al.* Pressure-Induced Phase Transformation, Reversible Amorphization, and Anomalous Visible Light Response in Organolead Bromide Perovskite. *J. Am. Chem. Soc.* **137**, 11144-11149, (2015).
- (6) Postorino, P. & Malavasi, L. Pressure-Induced Effects in Organic–Inorganic Hybrid Perovskites. *J. Phys. Chem. Lett.* **8**, 2613-2622, (2017).

Added Figure S13. TEM (a) images of the originally starting Cs_4PbBr_6 NCs. TEM (b) images of the decompressed Cs_4PbBr_6 NCs when the pressure was released to the ambient conditions.

Added Figure S14. Pressure-dependent PL spectra of Cs₄PbBr₆ NCs randomly dispersed in toluene.

In summary, following the kind suggestions and insightful comments of the reviewer, we carefully rechecked our manuscript and made some amendments to response to the reviewer's issues. Hopefully we have addressed all of your concerns. The corresponding revised details are highlighted in red and can be found in Line 7-15, Page 7; Line 27-30, Page 8; Line 1-16, Line 18-30, Page 9 and Line 9-23, Page 15 of the revised manuscript, as additional Experimental Section; Line 4-14, Page 15; Figure S3; Figure S13; Figure S14 and their captions in Supporting Information.

Dear Reviewer: 2

First of all, we acknowledge your comments and suggestions very much, which are valuable in improving the quality of our manuscript. We revised our manuscript in accordance you're your instructive guidance and we feel that the revised manuscript is a great improvement on the original.

***Comment:** The manuscript deals with pressure dependence of the structural and luminescence properties of Cs₄PbBr₆. There is novelty in this work and the results are quite interesting. In my opinion, the findings may be useful in obtaining an understanding of the intriguing luminescence properties of Cs₄PbBr₆. While I am quite positive about the manuscript and like to see it published in this journal, the authors need to address the following points and revise the manuscript appropriately before it can be recommended for publication.*

***Comment 1:** Why does the exciton binding energy increase with increase in pressure?*

Author reply: Thank you very much for the review's insightful comments. Generally, the exciton binding energy can be qualitatively analyzed using the following expression:³⁹

$$E_b = \frac{m^* e^4}{2\hbar^2 \mathcal{E}^2}$$

where m^* is the effective mass, e is the electric charge, \mathcal{E} is the dielectric constant, and \hbar is Planck constant. We can see that the exciton binding energy is proportional to the value of effective mass of m^* . In this study, monoclinic Cs₄PbBr₆ NCs also have the 0D perovskite structure that consists of the disconnected [PbBr₆]⁴⁻ octahedra isolated by Cs⁺ cations. This unique structure renders a strong quantum confinement to confine charge carriers inside the octahedra to form bound excitons.^{22,40} Upon further compression, the skeleton of [PbBr₆]⁴⁻ octahedra were persistently contracted, which further localized the charge carrier and increases their effective mass (m^*).⁴¹ Therefore, the exciton binding energy exhibited an increase with the decreased size of octahedra at high pressure.⁴²⁻⁴⁴ In addition, the exciton binding energy can be deduced from the energy difference between the band gap energy and exciton absorption peak. From the pressure-dependent absorption spectra of Cs₄PbBr₆ NCs, an increase in exciton binding energy with compression can also be observed. According to the reviewer's advice, we further make some revision and add necessary discussions in the revised manuscript.

The corresponding revised details are highlighted in red and can be found in Line 17-29, Page 11 of the revised manuscript.

REFERENCE

- (22) Saidaminov, M. I. *et al.* Pure Cs₄PbBr₆: Highly Luminescent Zero-Dimensional Perovskite Solids. *ACS Energy Lett.* **1**, 840-845, (2016).
- (39) Miyata, A. *et al.* Direct measurement of the exciton binding energy and effective masses for charge carriers in organic–inorganic tri-halide perovskites. *Nat. Phys.* **11**, 582, (2015).
- (40) Cha, J.-H. *et al.* Photoresponse of CsPbBr₃ and Cs₄PbBr₆ Perovskite Single Crystals. *J. Phys. Chem. Lett.* **8**, 565-570, (2017).
- (41) L., M. R., E., E. G., J., S. H., B., J. M. & M., H. L. Temperature - Dependent Charge - Carrier Dynamics in CH₃NH₃PbI₃ Perovskite Thin Films. *Adv. Funct. Mater.* **25**, 6218-6227, (2015).
- (42) Ramvall, P., Tanaka, S., Nomura, S., Riblet, P. & Aoyagi, Y. Observation of confinement-dependent exciton binding energy of GaN quantum dots. *Appl. Phys. Lett.* **73**, 1104-1106, (1998).
- (43) Meulenbergh, R. W. *et al.* Determination of the Exciton Binding Energy in CdSe Quantum Dots. *ACS Nano* **3**, 325-330, (2009).
- (44) Takagahara, T. & Takeda, K. Theory of the quantum confinement effect on excitons in quantum dots of indirect-gap materials. *Phys. Rev. B* **46**, 15578-15581, (1992).

Comment 2: *Why the VB and conduction band edge energy levels or self-trapped state energy level not affected by pressure?*

Author reply: We highly appreciate the reviewer for the constructive comments. In fact, the VB and CB edge energy levels or self-trapped state energy levels are affected by pressure. According to the advice, we further added additional discussions about the changes of energy levels under high pressure in the revised manuscript and supporting information.

The details are stated as follows:

The partial density of states indicated that for the structures of Cs₄PbBr₆ NCs at ambience and high pressure, the valence band maximum (VBM) was identified as the antibonding hybridization between the Pb 6s and Br 4p orbitals, whereas the conduction band minimum (CBM) state was governed by the strong nonbonding Pb 6p orbitals in the isolated [PbBr₆]⁴⁻ octahedra (Figure S2 and S8). In the range from 1 atm to 4.01 GPa, the [PbBr₆]⁴⁻ octahedra underwent a contraction with the increase in pressure (Figure S16b). The CBM is less-sensitive to the contraction of the Pb-Br bond length and slightly changed upon compression owing to the nonbonding characteristic of Pb 6p orbitals, whereas the Pb 6s orbital and Br 4p orbital became closer with the increasing pressure. This change leads to an increased degree of Pb-Br electron cloud overlap.

Hence, the VBM energy increases as a result of the enhanced coupling between the Pb $6s$ and the Br $4p$ orbitals, thereby ultimately cause the red shift of the band gap with pressure. As the Cs_4PbBr_6 perovskite was subjected to pressure beyond 4.01 GPa, the $[\text{PbBr}_6]^{4-}$ octahedra underwent a stark distortion to accommodate the Jahn-Teller effect, resulting in the reduced overlap between electron clouds of the Pb $6s$ and Br $4p$ (Figure S16c,d). The coupling of Pb $6s$ and Br $4p$ orbitals weakened, thus resulting in the decreasing energy of the VBM.⁷ In addition, the FWHM of PL was sharply decrease in the range from 3.01 GPa to 6.17 GPa (Figure S3c), indicating that the electron-phonon coupling became weak gradually.⁵ According to the Franck-Condon principle, the ΔQ can reflect the strength of electron-phonon coupling (Figure 4). The ΔQ was rationally reduced with the increasing pressure, indicating that the self-trapped states gradually approach to the bound excitonic states. Meanwhile, the reduction of ΔQ also leads to a reduction of the energy barrier between self-trapped state and bound excitonic state. Therefore, more bound excitons will ultimately converse to the self-trapped excitons by the pathway of B to C.

The corresponding revised details are highlighted in red and can be found in Line 6-21, Page 19 and Line 1-10, Page 20 of the revised Supporting Information.

REFERENCE

- (5) McCall, K. M., Stoumpos, C. C., Kostina, S. S., Kanatzidis, M. G. & Wessels, B. W. Strong Electron-Phonon Coupling and Self-Trapped Excitons in the Defect Halide Perovskites $\text{A}_3\text{M}_2\text{I}_9$ (A = Cs, Rb; M = Bi, Sb). *Chem. Mater.* **29**, 4129-4145, (2017).
- (7) Kong, L. *et al.* Simultaneous band-gap narrowing and carrier-lifetime prolongation of organic-inorganic trihalide perovskites. *Proc. Natl. Acad. Sci. U. S. A.* **113**, 8910-8915, (2016).

Revised Figure S16. (a) The band gaps against pressures for Phase I and Phase II of Cs₄PbBr₆ NCs in situ measured in a DAC apparatus. (b, c, d) Schematic illustrations of the band gap changes governed by the contraction and distortion of [PbBr₆]⁴⁻ octahedra before and after structural phase transition.

Revised Figure 4. Configuration coordinate model of PL enhancement for the high-pressure monoclinic phase of Cs₄PbBr₆ NCs at 4 GPa (a) and 6 GPa (b). Therein, the absorption transition upon excitation is described from A to B. The STEs recombination emission is depicted from C to D. Two distinct paths at 4 GPa are represented: thermal overcoming of the barrier (1) and tunneling (2). (FC): free carrier state, (BE): bound exciton state, ΔQ: the equilibrium position difference between ground and self-trapped exciton state.

Added Figure S3. (a) Normalized PL intensity as a function of pressure. (b) The PL location of Cs₄PbBr₆ NCs against pressures. (c) Full width at half-maximum (FWHM) vs pressure.

Comment 3: As the self-trapped state is of specific energy, the broad nature of the emission is not understandable.

Author reply: We are very sorry for confusing the reviewer due to our unclear statements. As the electron-phonon coupling plays an important role in determining the bandwidth of the PL and the Stokes shift. The strong electron-phonon coupling would lead to the localized charge and emission of phonons that alter the energy of the emitted photon. The broad emission was closely related to the multiple radiative transitions accompanied by the phonons emitted as the electron relaxes from points D to A (Figure 4). These transitions have equidistant spacing given by the phonon energy E_{ph} , the n th transition involving n phonons has energy $E_n = E - nE_{ph}$ ($E = E_C - E_A$ is known as the zero-phonon line of the emission band)

According to the reviewers' valuable enlightenments, we have modified the schematic illustrations as configuration coordinate diagram to make the mechanism of the emission enhancement more clear, as shown in Figure 4.

The revised details are stated as follows:

In this process, the electron-phonon coupling plays an important role in determining the bandwidth of the PL and Stokes shift.²⁵ The broad emission consists of several radiative transitions (from C to D) accompanied by the different extent of phonons relaxation (from D to A).

The corresponding revised details are highlighted in red and blue and can be found in Line 3-6, Page 13 of the revised manuscript.

REFERENCE

(25) McCall, K. M., Stoumpos, C. C., Kostina, S. S., Kanatzidis, M. G. & Wessels, B. W. Strong Electron-Phonon Coupling and Self-Trapped Excitons in the Defect Halide Perovskites $A_3M_2I_9$ ($A = Cs, Rb$; $M = Bi, Sb$). *Chem. Mater.* **29**, 4129-4145, (2017).

Revised Figure 4. Configuration coordinate model of PL enhancement for the high-pressure monoclinic phase of Cs_4PbBr_6 NCs at 4 GPa (a) and 6 GPa (b). Therein, the absorption transition upon excitation is described from A to B. The STEs recombination emission is depicted from C to D. Two distinct paths at 4 GPa are represented: thermal overcoming of the barrier (1) and tunneling (2). (FC): free carrier state, (BE): bound exciton state, ΔQ : the equilibrium position difference between ground and self-trapped exciton state.

Comment 4: Insight into the formation ‘self-trapped state’ due to pressure-induced distortion of the octahedron is lacking

Author reply: We are grateful for the reviewer’s expertise and kind suggestions. According to the advice, we have introduced more descriptions about the formation self-trapped state within monoclinic Cs_4PbBr_6 NCs from the experiments and theoretical calculations. Some necessary references were also added.

The revised details are stated as follows:

STEs are strongly dependent on the dimensionality of the system, with stronger self-trapping in lower-dimensional materials.²⁵ In addition, such STE recombination is highly dependent upon electron-phonon coupling since electron-phonon coupling causes local lattice distortions, polarons, which bind the charge carriers to form STEs.²⁶ STEs are generally observed in distorted crystal structures.

In situ high-pressure synchrotron ADXRD patterns of the Cs₄PbBr₆ NCs were collected as shown in Figure S4. The evolution of ADXRD diffraction patterns demonstrates that a structural phase transition from rhombohedral (Phase I) to monoclinic (Phase II) structure occurs at 3.04 GPa and ends at 4.01 GPa, accompanied by a significant distortion of [PbBr₆]⁴⁻ octahedra to accommodate the Jahn-Teller effect (Figure S9). The distorted octahedra produce small polarons that locally trap charge carriers, resulting in STEs. Therein, the pressure plays a critical role in inducing the distortion of octahedra within Cs₄PbBr₆ NCs. In order to better understand the formation of STEs, we calculated the high-pressure electronic band structure of Cs₄PbBr₆ by DFT using HSE including spin-orbit coupling (Figure S10a).²⁷ It is found that the static lattice bands (black solid lines) happen to be distorted in the presence of the phonon (red dashed lines), indicating a strong electron-phonon coupling within the monoclinic structure.²⁸ In addition, a marked change in relative strength of Raman modes is an indicator of strong coupling between the excited electrons and Raman active vibrations of the lattice. Indeed, we observe an intense relative change of Raman scattering modes in Cs₄PbBr₆ NCs on increasing pressure, further confirming the strong electron-phonon coupling in its high-pressure structure (Figure S11).²⁵ We have also investigated the possible presence of polaronic states (Figure S10b). The large mismatch of the total density of states between the ideal structure and STE structure further corroborates the view that polarons may be stabilized during structural optimization. The radiative recombination of STEs gives rise to a class of mid-band-gap emission associated with the Jahn-Teller distortion. Hence, the high-pressure emission has nothing to do with the variation of the intrinsic band gaps. Similar emission has been reported on organic-inorganic perovskites with corrugated bilayer structures and in 0D all-inorganic perovskites, A₃M₂I₉ (A=Cs, Rb and M= Sb, Bi).^{25,29,30} The broad emission reported for these materials was also attributed to the recombination of STEs because of the lattice distortion. Note that as the pressure increases in the stability field of the initial rhombohedral phase, the strong electrostatic interaction makes the six bond lengths of Pb-Br smaller within the regular [PbBr₆]⁴⁻ octahedra (Figure S12). During this process the octahedra do not distort (Table S1), and thus no PL response is seen below 3.01 GPa.

The revised details are highlighted in red and can be found in Line 19-30, Page 7 and Line 1-26, Page 8 in the revised manuscript.

REFERENCE

- (25) McCall, K. M., Stoumpos, C. C., Kostina, S. S., Kanatzidis, M. G. & Wessels, B. W. Strong Electron-Phonon Coupling and Self-Trapped Excitons in the Defect Halide Perovskites $A_3M_2I_9$ ($A = Cs, Rb$; $M = Bi, Sb$). *Chem. Mater.* **29**, 4129-4145, (2017).
- (26) Stoneham, A. M. *et al.* Trapping, self-trapping and the polaron family. *J. Phys.: Condens. Matter* **19**, 255208 (2007).
- (27) Kang, B. & Biswas, K. Exploring Polaronic, Excitonic Structures and Luminescence in $Cs_4PbBr_6/CsPbBr_3$. *J. Phys. Chem. Lett.* **9**, 830-836, (2018).
- (28) Monserrat, B., Dreyer, C. E. & Rabe, K. M. Phonon-assisted optical absorption in $BaSnO_3$ from first principles. *Phys. Rev. B* **97**, 104310, (2018).
- (29) Dohner, E. R., Jaffe, A., Bradshaw, L. R. & Karunadasa, H. I. Intrinsic White-Light Emission from Layered Hybrid Perovskites. *J. Am. Chem. Soc.* **136**, 13154-13157, (2014).
- (30) Dohner, E. R., Hoke, E. T. & Karunadasa, H. I. Self-Assembly of Broadband White-Light Emitters. *J. Am. Chem. Soc.* **136**, 1718-1721, (2014).

Figure S9. Crystal structures and structure units of Cs_4PbBr_6 NCs at 1 atm (a, d), 4.01 GPa (b, e) and 6.12 GPa (c, f), respectively.

Added Figure S10. (a) Electronic band structure of Cs_4PbBr_6 with and without the phonon is calculated by DFT using HSE hybrid functions. The static lattice bands (black solid lines) shift in the presence of the phonon (red dashed lines). (b) Total density of state of Cs_4PbBr_6 in ideal structure or STE structure.

Revised Figure S11. (a) Selected Raman spectra of Cs_4PbBr_6 NCs at elevated pressure. (b) Raman spectra of Cs_4PbBr_6 NCs at 0.45 GPa. (c) Raman spectra of Cs_4PbBr_6 NCs at 4.08 GPa.

Added Figure S12. Schematic diagram of the three non-equivalent Pb-Br bonds and two non-equivalent Br-Pb-Br angles within isolated $[\text{PbBr}_6]^{4-}$ octahedra for monoclinic Cs_4PbBr_6 crystals.

Added Table S1. The three Pb-Br bonds and two Br-Pb-Br angles within isolated $[\text{PbBr}_6]^{4-}$ octahedra for Cs_4PbBr_6 crystal as function of pressure.

	L_1 (Å)	L_2 (Å)	L_3 (Å)	A_1 (degrees)	A_2 (degrees)
1 atm	2.989	2.989	2.989	90	90
2.01 GPa	2.925	2.925	2.925	90	90
4.01 GPa	2.891	2.767	2.845	86.043	90.578
5.01 GPa	2.89	2.766	2.852	85.824	90.292
6.12 GPa	2.881	2.761	2.856	85.506	90.142

Comment 5: Why do the excited carriers bind readily to the free exciton state when the nanocrystals are in their monoclinic phase (sentence 218-220)?

Author reply: We are very sorry for confusing the reviewer due to our unclear statements. The excited carriers bind readily to the free exciton state when the nanocrystals are in their monoclinic phase. This phenomenon is ascribed to the strong exciton binding energy of the crystals, originating from their unique crystal structure. The monoclinic phase Cs_4PbBr_6 NCs also has have the 0D perovskite structure that consists of the disconnected $[\text{PbBr}_6]^{4-}$ octahedra isolated by Cs^+ cations, possessing exhibiting strong quantum confinement of charge carriers. Thus it is difficult for the excitons to flow across the internal crystals, and they can be bound because of the high exciton binding energy.^{22,40} On the other hand, compared with ambient structure, the skeleton of $[\text{PbBr}_6]^{4-}$ octahedra was persistently contracted in

monoclinic phase, which further localized charge carrier, thus and increase increasing their effective mass. Generally, the exciton binding energy can be qualitatively analyzed using the following expression:³⁹

$$E_b = \frac{m^* e^4}{2\hbar^2 \epsilon^2}$$

where m^* is the effective mass, e is the electric charge, ϵ is the dielectric constant, and \hbar is Planck constant. We can see that the exciton binding energy is proportional to the value of effective mass. Therefore, the exciton binding energy can be increased due to the increase in the effective mass of carriers within the monoclinic structure of Cs₄PbBr₆ NCs. Thereby, it can be easily concluded that the excited carriers can be readily localized to the bound exciton state when the Cs₄PbBr₆ NCs are lied in their high-pressure monoclinic phase.

The revised details are highlighted in red and can be found in Line 22-29, Page 11 in the revised manuscript.

REFERENCE

- (22) Saidaminov, M. I. *et al.* Pure Cs₄PbBr₆: Highly Luminescent Zero-Dimensional Perovskite Solids. *ACS Energy Lett.* **1**, 840-845, (2016).
- (39) Miyata, A. *et al.* Direct measurement of the exciton binding energy and effective masses for charge carriers in organic–inorganic tri-halide perovskites. *Nat. Phys.* **11**, 582, (2015).
- (40) Cha, J.-H. *et al.* Photoresponse of CsPbBr₃ and Cs₄PbBr₆ Perovskite Single Crystals. *J. Phys. Chem. Lett.* **8**, 565-570, (2017).

Comment 6: *Minor points: There is a number of grammatical and typographical errors which are to be corrected.*

Author reply: Thanks very much for the reviewer's professional knowledge and great patience to spot the mistakes, and we are so sorry for our poor English expressions to confuse the reviewer. To this end, we invoked Prof. Simon A. T. Redfern, who is a high-pressure scientist at University of Cambridge, to make essential editing on the language and fruitful discussions. Therefore, the writing is a great improvement on the original. Meanwhile, Prof. Simon A. T. Redfern is also added as one of co-authors of the manuscript due to the significant contribution in improving the quality of the manuscript. We hope this English editing will help the readers to better understand this paper.

In summary, following the kind suggestions and insightful comments of the reviewer, we carefully rechecked our manuscript and made some amendments to response to the reviewer's issues. Hopefully we have addressed all of your concerns. The corresponding revised details are highlighted in red and can be found in Line 19-30, Page 7; Line 1-26, Page 8; Line 17-29, Page 11; Line 3-6, Page 13 and revised Figure 4 in the revised manuscript; Line 6-21, Page 19; Line 1-10, Page 20; Figure S10; Figure S11; Figure S12; Table S1; Figure S16 and their captions in the revised Supporting Information.

Dear Reviewer: 3

First of all, we acknowledge your comments and suggestions very much, which are valuable in improving the quality of our manuscript. We revised our manuscript in accordance with your instructive guidance and we feel that the revised manuscript is a great improvement on the original.

***Comment:** This paper reports on the pressure induced emission of halide perovskite Cs₄PbBr₆ nanocrystals (0D NCs). The authors show that films of non-emitting 0D NCs can emit once they are exposed to high pressure up to 3.01 GPa. The authors attribute such emission to self-trapped electrons from highly distorted [PbBr₆]⁴⁻ octahedra at high pressures.*

This paper is surely interesting and might be of potential interest for a wide audience. On the other hand, considering the ongoing debate about the origin of the emission for these systems, the authors provide very little evidence about what produces such emission. In addition, the explanation of the emission mechanism is rather confusing and not clear. In short, the authors do not provide sufficient and compelling explanation on the origin of the emission of these systems at high pressure. To make this work publishable, the authors should be less vague on their claims and be able to prove what they speculate about.

***Comment 1:** From line 90-95, the authors attribute the mid-gap state emission in 0D NC (at 1 atm) to state derived from halogen vacancy. This is actually not true. The halogen vacancy is one of the many theories that explain emission, and very likely the least accurate, considering the high presence of halogen in this system (more than 6:1 vs Pb). So far, the more robust explanation for the emission is attributed to the presence of 3D perovskite impurities embedded inside the Cs₄PbBr₆ matrix. See, for example, Manna's papers for details. This explains also why emitting 0D powders absorbs at around 500nm at energies similar to the emission.*

Author reply: We highly appreciate the reviewer for the insightful comments. The synthesized Cs₄PbBr₆ NCs exhibited no any emission at ambient condition, which is in good accordance with the previously well-established reports.^{18,19} Several groups reported a strong green luminescence in Cs₄PbBr₆ powders, single-crystals, and NCs.²⁰⁻²² There remains ongoing debate about the origin of the emission for these systems. Some works believed that the origin of the emission can be attributed to the mid-band gap states formed because of the halogen vacancy. However, the recent report by Manna *et al.*²³ proposed a more reasonable explanation, deepening insight into the non-fluorescent mechanism of the Cs₄PbBr₆ systems. According to their reports,^{8,23} Cs₄PbBr₆ exhibits a Br:Pb ratio that is usually higher than 6, thus implying that Cs₄PbBr₆ are rather halide rich. The halogen vacancy is very likely the least accurate. In addition, if the green emission was to originate from a midgap state or a trap state, then the absorption of the Cs₄PbBr₆ host

would remain unaltered, i.e., it would have no absorption features in the visible range. However, in the case of the green luminescent Cs₄PbBr₆, the absorption and PL spectra show features in the visible range (around 510-520 nm), which is in the same range as the green emission. The green emission in the Cs₄PbBr₆ is therefore often reported with a very small Stokes shift of only 28 meV,²⁰⁻²² suggesting that the green emission originates from a direct band-band transition in the visible, rather than from emission from a midgap defect state. This conclusion is completely contradiction with the emission from the halogen vacancy. As also pointed out by the reviewer, the more robust explanation for the emission is attributed to the presence of 3D perovskite impurities embedded inside the Cs₄PbBr₆ matrix. Recently, Manna and co-workers have also reported that the green emission in Cs₄PbBr₆ originates from CsPbBr₃ impurities encapsulated in the Cs₄PbBr₆ bulk matrix.^{8,23} This indeed explains why those emitting 0D powders absorb at around 500 nm at energies similar to that of the emission. According to the advice, we have revised the explanation about the origin of emission in our manuscript.

The revised details are stated as follows:

In addition, the steady-state PL of samples was performed at ambient condition and no emission was observed, matching with previous reports on bulk Cs₄PbBr₆ powders and films.^{18,19} Although some works showed that Cs₄PbBr₆ NCs or the bulk structure possessed fluorescence in the visible region,²⁰⁻²² there remains ongoing debate about the origin of the emission for these systems. The recent report by Manna *et al.*²³ proposed a more reasonable explanation, deepening insight into the non-fluorescent mechanism of the Cs₄PbBr₆ systems. According to their reports,^{8,23} the origin of PL can be attributed to the presence of 3D perovskite impurities embedded inside the Cs₄PbBr₆ matrix, rather than the intrinsic emission of Cs₄PbBr₆.

The revised details are highlighted in red and can be found in **Line 20-29, Page 5** of the revised manuscript.

REFERENCE

- (8) Akkerman, Q. A. *et al.* Nearly Monodisperse Insulator Cs₄PbX₆ (X = Cl, Br, I) Nanocrystals, Their Mixed Halide Compositions, and Their Transformation into CsPbX₃ Nanocrystals. *Nano Lett.* **17**, 1924-1930, (2017).
- (18) Nikl, M. *et al.* Photoluminescence of Cs₄PbBr₆ crystals and thin films. *Chem. Phys. Lett.* **306**, 280-284, (1999).
- (19) Kondo, S., Amaya, K. & Saito, T. Localized optical absorption in Cs₄PbBr₆. *J. Phys.: Condens. Matter* **14**, 2093 (2002).

- (20) Zhang, Y. *et al.* Zero-Dimensional Cs₄PbBr₆ Perovskite Nanocrystals. *J. Phys. Chem. Lett.* **8**, 961-965, (2017).
- (21) Seth, S. & Samanta, A. Fluorescent Phase-Pure Zero-Dimensional Perovskite-Related Cs₄PbBr₆ Microdisks: Synthesis and Single-Particle Imaging Study. *J. Phys. Chem. Lett.* **8**, 4461-4467, (2017).
- (22) Saidaminov, M. I. *et al.* Pure Cs₄PbBr₆: Highly Luminescent Zero-Dimensional Perovskite Solids. *ACS Energy Lett.* **1**, 840-845, (2016).
- (23) Akkerman, Q. A., Abdelhady, A. L. & Manna, L. Zero-Dimensional Cesium Lead Halides: History, Properties, and Challenges. *J. Phys. Chem. Lett.* **9**, 2326-2337, (2018).

Comment 2: *In lines 52-55, the authors say that the emission is related to the Jahn-Teller emission due to distorted octahedral. However, later in the manuscript and in the supporting info, the authors show, using DFT calculations, that the bandgap is reduced from 3.23 eV to 3.14 eV, only 0.09 eV, therefore indicating that despite the distortion, the change in energy is very minor.*

Author reply: We are very sorry for confusing the reviewer due to our unclear statements. In fact, the origin of broadening emission in this study was attributed to the radiative recombination of self-trapped excitons (STEs), a representative mid-band-gap emission, rather than an emission from band-gap transition. The Cs₄PbBr₆ NCs at 6.23 GPa exhibited a broad emission, with a maximum intensity centered at 549 nm and a large full width at half maximum (FWHM) of ~ 150 nm. Meanwhile, the absorption peak located at 326 nm, indicating a very large Stokes shift exceeding 220 nm. Our explanation for the emission of Cs₄PbBr₆ NCs at high pressure is recombination between the STEs. Such STE recombination is highly dependent upon electron-phonon coupling since electron-phonon coupling causes local lattice distortions, polarons, which bind the charge carriers to form STEs.²⁶ STEs are generally observed in distorted crystal structures. STEs were generally observed in distorted crystals, and the electron-phonon coupling is a main factor for the large bandwidth and Stokes shift of the emission. In our investigation, the pressure plays an critical role in inducing the distortion of octahedra within Cs₄PbBr₆ NCs. The distorted octahedra generate a completely new recombination pathway of excitons via the radiative recombination of STEs, which is independent of the behavior of band gaps. This conclusion also rationalizes the very large Stokes shift between the emission location and the band-gap edge. We also calculated the high-pressure electronic band structure of Cs₄PbBr₆ with and without the phonon by DFT using HSE including spin-orbit coupling (Figure S10a).²⁸ It is found that the static lattice bands (black solid lines) happen to be distorted in the presence of the phonon (red dashed lines), indicating a strong electron-phonon coupling within high-pressure monoclinic structure.²⁹ Recently, Biswas and his coworkers have also reported the polaronic and excitonic features in

the Cs_4PbBr_6 by using hybrid density functional calculations.²⁷ This conclusion is well agreed with our analysis. Note that the broad emission is highly related to the pressure-induced distorted octahedra. As the pressure increased in the initial rhombohedral phase range, although the strong electrostatic interaction decreased the six bond lengths of Pb-Br within the regular $[\text{PbBr}_6]^{4-}$ octahedra gradually (Figure S12, Table S1), the distorted octahedra cannot be observed in this process, and thus no PL response to the external pressure occurred below 3.01 GPa.

Moreover, the partial density of states, as shown in Figure S2 and S8, suggested that for Cs_4PbBr_6 perovskites whether at ambience or high pressure, the valence band maximum (VBM) was identified as the antibonding hybridization between the Pb $6s$ and Br $4p$ orbitals, whereas the conduction band minimum (CBM) state was governed by the strong nonbonding Pb $6p$ orbitals in the isolated $[\text{PbBr}_6]^{4-}$ octahedra. The unique separated octahedral structure in Cs_4PbBr_6 would reduce the interactions of Pb $6s$ and Br $4p$.²⁸ Therefore, the strength of coupling between the Pb $6s$ and the Br $4p$ orbitals exhibited a ignorable increase, although the $[\text{PbBr}_6]^{4-}$ octahedra underwent a contraction with increasing pressure. This accordingly explains why the change in energy is very minor despite the distortion.

The revised details are stated as follows:

The profile of the emission under high-pressure conditions appears asymmetric and skewed to the low-energy side with a very large Stokes shift exceeding 220 nm. We can understand the emission in Cs_4PbBr_6 NCs in terms of recombination between self-trapped excitons (STEs), a class of emission from mid-band-gap states. STEs are strongly dependent on the dimensionality of the system, with stronger self-trapping in lower-dimensional materials.²⁵ In addition, such STE recombination is highly dependent upon electron-phonon coupling since electron-phonon coupling causes local lattice distortions, polarons, which bind the charge carriers to form STEs.²⁶ STEs are generally observed in distorted crystal structures.

In situ high-pressure synchrotron ADXRD patterns of the Cs_4PbBr_6 NCs were collected as shown in Figure S4. The evolution of ADXRD diffraction patterns demonstrates that a structural phase transition from rhombohedral (Phase I) to monoclinic (Phase II) structure occurs at 3.04 GPa and ends at 4.01 GPa, accompanied by a significant distortion of $[\text{PbBr}_6]^{4-}$ octahedra to accommodate the Jahn-Teller effect (Figure S9). The distorted octahedra produce small polarons that locally trap charge carriers, resulting in STEs. Therein, the pressure plays a critical role in inducing the distortion of octahedra within Cs_4PbBr_6 NCs. In order to better understand the formation of STEs, we calculated the high-pressure electronic band structure of Cs_4PbBr_6 by DFT using HSE including spin-orbit coupling (Figure S10a).²⁷ It is found that the static lattice bands (black solid lines) happen to be distorted in the presence of the phonon (red dashed lines),

indicating a strong electron-phonon coupling within the monoclinic structure.²⁸ In addition, a marked change in relative strength of Raman modes is an indicator of strong coupling between the excited electrons and Raman active vibrations of the lattice. Indeed, we observe an intense relative change of Raman scattering modes in Cs₄PbBr₆ NCs on increasing pressure, further confirming the strong electron-phonon coupling in its high-pressure structure (Figure S11).²⁵ We have also investigated the possible presence of polaronic states (Figure S10b). The large mismatch of the total density of states between the ideal structure and STE structure further corroborates the view that polarons may be stabilized during structural optimization. The radiative recombination of STEs gives rise to a class of mid-band-gap emission associated with the Jahn-Teller distortion. Hence, the high-pressure emission has nothing to do with the variation of the intrinsic band gaps. Similar emission has been reported on organic-inorganic perovskites with corrugated bilayer structures and in 0D all-inorganic perovskites, A₃M₂I₉ (A=Cs, Rb and M= Sb, Bi).^{25,29,30} The broad emission reported for these materials was also attributed to the recombination of STEs because of the lattice distortion. Note that as the pressure increases in the stability field of the initial rhombohedral phase, the strong electrostatic interaction makes the six bond lengths of Pb-Br smaller within the regular [PbBr₆]⁴⁻ octahedra (Figure S12). During this process the octahedra do not distort (Table S1), and thus no PL response is seen below 3.01 GPa.

The revised details are highlighted in red and can be found in Line 16-30, Page 7 and Line 1-26, Page 8 of the revised manuscript.

REFERENCE

- (25) McCall, K. M., Stoumpos, C. C., Kostina, S. S., Kanatzidis, M. G. & Wessels, B. W. Strong Electron-Phonon Coupling and Self-Trapped Excitons in the Defect Halide Perovskites A₃M₂I₉ (A = Cs, Rb; M = Bi, Sb). *Chem. Mater.* **29**, 4129-4145, (2017).
- (26) Stoneham, A. M. *et al.* Trapping, self-trapping and the polaron family. *J. Phys.: Condens. Matter* **19**, 255208 (2007).
- (27) Kang, B. & Biswas, K. Exploring Polaronic, Excitonic Structures and Luminescence in Cs₄PbBr₆/CsPbBr₃. *J. Phys. Chem. Lett.* **9**, 830-836, (2018).
- (28) Monserrat, B., Dreyer, C. E. & Rabe, K. M. Phonon-assisted optical absorption in BaSnO₃ from first principles. *Phys. Rev. B* **97**, 104310, (2018).
- (29) Dohner, E. R., Jaffe, A., Bradshaw, L. R. & Karunadasa, H. I. Intrinsic White-Light Emission from Layered Hybrid Perovskites. *J. Am. Chem. Soc.* **136**, 13154-13157, (2014).
- (30) Dohner, E. R., Hoke, E. T. & Karunadasa, H. I. Self-Assembly of Broadband White-Light Emitters. *J. Am. Chem. Soc.* **136**, 1718-1721, (2014).

Added Figure S10. (a) Electronic band structure of Cs₄PbBr₆ with and without the phonon is calculated by DFT using HSE hybrid functionals. The static lattice bands (black solid lines) shift in the presence of the phonon (red dashed lines). (b) Total density of state of Cs₄PbBr₆ in ideal structure or STE structure.

Added Figure S12. Schematic diagram of the three non-equivalent Pb-Br bonds and two non-equivalent Br-Pb-Br angles within isolated $[\text{PbBr}_6]^{4-}$ octahedra for monoclinic Cs_4PbBr_6 crystals.

Added Table S1. The three Pb-Br bonds and two Br-Pb-Br angles within isolated $[\text{PbBr}_6]^{4-}$ octahedra for Cs_4PbBr_6 crystal as function of pressure.

	L_1 (Å)	L_2 (Å)	L_3 (Å)	A_1 (degrees)	A_2 (degrees)
1 atm	2.989	2.989	2.989	90	90
2.01 GPa	2.925	2.925	2.925	90	90
4.01 GPa	2.891	2.767	2.845	86.043	90.578
5.01 GPa	2.89	2.766	2.852	85.824	90.292
6.12 GPa	2.881	2.761	2.856	85.506	90.142

Comment 3: In the caption of Figure 2, the authors mention absorption and PL emission, however the graphs seem to be only related to emission.

Author reply: Thanks very much for the reviewer's carefulness, and we are very sorry for our mistakes. In the revised manuscript, we have removed the words '*optical absorptions and*' from the caption of Figure 2.

The revised details are stated as follows:

Figure 2. (a), (b) and (c) Changes in the PL spectra of Cs₄PbBr₆ NCs under pressure. Black arrows indicate the evolution of the PL spectra as a function of pressure. (d) Pressure-dependent Chromaticity coordinates of the emissions.

The revised details are highlighted in red and can be found in Line 3-5, Page 6 of the revised manuscript.

Comment 4: *From line 125-129, the authors analyze the Raman spectra to explain the behavior of low vibrational mode. While this is per se a good idea, the whole explanation is quite confusing and not sufficiently clear. There is lots of wording, but not clear insights. Especially the sentence “local fluctuation ... will induce the formation of small polarons, resulting in trapping of excitons by the lattice”. What does it mean? Are you actually proving that these polarons are formed and they have consequences on trapping? This seems to me just speculation and not a concrete analysis.*

Author reply: Thanks very much for the reviewer’s insightful comments, and we are very sorry for confusing the reviewer due to our unclear statements. The investigations of the vibrational properties of Cs₄PbBr₆ NCs are important to explore the electron-phonon interaction, which plays a significant role in the radiative recombination of STEs. The marked change in relative strength of Raman modes is an indicator of strong coupling between the excited electrons and Raman active vibrations of the lattice. Raman spectroscopy detects the dipole-dipole polarizability, so the strong scattering interactions reflect the great lattice’s proclivity to local fluctuations. This propensity to form local charges then plays a role in the charge transport. The interaction between the local charges and free charge carriers will cause the lattice to deform.⁵ These deformations travel with the carriers, forming polarons that bind charge carriers to form STEs and increase the effective mass of charge carriers. Indeed, we observe an intense relative change of Raman scattering modes in Cs₄PbBr₆ NCs, further confirming the strong electron-phonon coupling in its high-pressure structure (Figure S11). Therefore, polarons and STEs may occur in the deformed octahedra. In order to confirm the formation of polarons, we investigated the possibility of such polaronic states using HSE including spin-orbit coupling in high-pressure structure of Cs₄PbBr₆ NCs (Figure S10b). The large mismatch of the total density of states between the ideal structure and STE structure further corroborates that the polarons can be stable during the process of structural optimization. Recently, Biswas and his coworkers have also confirmed that the 0D Cs₄PbBr₆ possesses polaronic and excitonic features by using hybrid density functional calculations,²⁷ which is well consistent with our results.

According to the reviewer’s advice, we further give the discussions on interpreting the high-pressure Raman experiments of Cs₄PbBr₆ NCs. We have also made some amendments and added some necessary references in our revised manuscript and Supporting Information. The revised details are stated as follows:

The evolution of Raman spectra ranging from 70 to 300 cm⁻¹ was shown in Figure S11. Three lattice

modes at 77, 88 and 127 cm^{-1} are observed at ambient conditions. These strong modes are associated with the vibrational modes of $[\text{PbBr}_6]^{4-}$ octahedra.⁴ As the pressure increased to 3.10 GPa, the two lattice modes in the 70-100 cm^{-1} region become so weak, while the relative intensity of lattice mode at 127 cm^{-1} is enhanced dramatically, consistent with the result of ADXRD. The three modes undergo a redistribution of intensities and remain stable beyond 4.08 GPa, reflecting that the structure of octahedra happened to be changed due to the phase transition. This phenomenon is also well agreed with the result of Rietveld refinements. Furthermore, Raman spectroscopy detects the dipole-dipole polarizability, so the strong scattering interactions reflect the great lattice's proclivity to local fluctuations. This propensity to form local charges then plays a role in the charge transport. The interaction between the local charges and free charge carriers will cause the lattice to deform.⁵ These deformations travel with the carriers, forming polarons that bind charge carriers to form STEs. Indeed, we observe an intense relative change of Raman scattering modes in Cs_4PbBr_6 NCs, further confirming the strong electron-phonon coupling in its high-pressure structure (Figure S11). Therefore, the polarons and STEs will appear in the deformed octahedra for the high-pressure structure of Cs_4PbBr_6 NCs.

In order to better understand the formation of STEs, we calculated the high-pressure electronic band structure of Cs_4PbBr_6 by DFT using HSE including spin-orbit coupling (Figure S10a).²⁷ It is found that the static lattice bands (black solid lines) happen to be distorted in the presence of the phonon (red dashed lines), indicating a strong electron-phonon coupling within the monoclinic structure.²⁸ In addition, a marked change in relative strength of Raman modes is an indicator of strong coupling between the excited electrons and Raman active vibrations of the lattice. Indeed, we observe an intense relative change of Raman scattering modes in Cs_4PbBr_6 NCs on increasing pressure, further confirming the strong electron-phonon coupling in its high-pressure structure (Figure S11).²⁵ We have also investigated the possible presence of polaronic states (Figure S10b). The large mismatch of the total density of states between the ideal structure and STE structure further corroborates the view that polarons may be stabilized during structural optimization.

The revised details are highlighted in red and can be found in Line 2-15, Page 8 of the revised manuscript, and in Line 4-16, Page 12; Line 1-6, Page 13; Figure S10; Figure S11 and their captions of the revised Supporting Information.

REFERENCE

(4) Cha, J.-H. *et al.* Photoresponse of CsPbBr_3 and Cs_4PbBr_6 Perovskite Single Crystals. *J. Phys. Chem. Lett.* **8**, 565-570, (2017).

(5) McCall, K. M., Stoumpos, C. C., Kostina, S. S., Kanatzidis, M. G. & Wessels, B. W. Strong Electron-Phonon Coupling and Self-Trapped Excitons in the Defect Halide Perovskites $A_3M_2I_9$ ($A = Cs, Rb$; $M = Bi, Sb$). *Chem. Mater.* **29**, 4129-4145, (2017).

(25) McCall, K. M., Stoumpos, C. C., Kostina, S. S., Kanatzidis, M. G. & Wessels, B. W. Strong Electron-Phonon Coupling and Self-Trapped Excitons in the Defect Halide Perovskites $A_3M_2I_9$ ($A = Cs, Rb$; $M = Bi, Sb$). *Chem. Mater.* **29**, 4129-4145, (2017).

(27) Kang, B. & Biswas, K. Exploring Polaronic, Excitonic Structures and Luminescence in $Cs_4PbBr_6/CsPbBr_3$. *J. Phys. Chem. Lett.* **9**, 830-836, (2018).

(28) Monserrat, B., Dreyer, C. E. & Rabe, K. M. Phonon-assisted optical absorption in $BaSnO_3$ from first principles. *Phys. Rev. B* **97**, 104310, (2018).

Added Figure S10. (a) Electronic band structure of Cs_4PbBr_6 with and without the phonon is calculated by DFT using HSE hybrid functionals. The static lattice bands (black solid lines) shift in the presence of the phonon (red dashed lines). (b) Total density of state of Cs_4PbBr_6 in ideal structure or STE structure.

Revised Figure S11. (a) Selected Raman spectra of Cs₄PbBr₆ NCs at elevated pressure. (b) Raman spectra of Cs₄PbBr₆ NCs at 0.49 GPa. (c) Raman spectra of Cs₄PbBr₆ NCs at 4.08 GPa.

Comment 5: In lines 132-134, the authors say that the electron-phonon coupling give rise to a highly distorted excited state. This sentence however is not backed-up by neither the experiments nor theory (no calculations have been done to look at this effect). On the other hand, a large Stokes shift might be a better and more reasonable explanation for the emission. The Jahn-Teller effect that would occur in the excited state might be indeed quite large. However, the fact that 0D NC are non-emissive works against this explanation. And DFT calculations show that the Jahn-Teller distortion is around ~ 1 eV, not enough to exhibit emission in the ~ 520 nm region. Additionally, the authors should prove that at high pressure such emission becomes optically active in terms of transition dipole moments (demonstrated by DFT calculations for example). In short, why without pressure, there is no emission, while with pressure there is emission if the Jahn-Teller is responsible to the emission?

Author reply: We highly appreciate the reviewer for the constructive comments, and we are so sorry for confusing the reviewer due to our unclear statements. In fact, the origin of broadening emission in our investigation was attributed to the radiative recombination of self-trapped excitons (STEs) in mid-band-gap state, rather than an emission from band-gap transition. The distorted excited state in our original manuscript

is thought to the self-trapping states. The STE recombination depends strongly on the electron-phonon coupling. The strong electron-phonon coupling will cause local distortions of the lattice, known as polarons, which bind the charge carriers to form STEs.²⁶ Therefore, the STEs were generally observed in distorted crystals, and the electron-phonon coupling is a main factor for the large bandwidth and Stokes shift of the emission. In our investigation, the pressure plays an important role in inducing the distortion of octahedra within Cs₄PbBr₆ NCs. The in situ high pressure synchrotron ADXRD patterns of the Cs₄PbBr₆ NCs indicated that in the range from 3.01 to 4.01 GPa, the NCs underwent a structural phase transition from rhombohedral to monoclinic structure. Note that the phase transition resulted in significantly distorted [PbBr₆]⁴⁻ octahedra to accommodate the Jahn-Teller effect. The distorted octahedra generate a completely new recombination pathway of excitons via the radiative recombination of STEs, which is independent of the behavior of band gaps. This conclusion also rationalizes the very large Stokes shift between the emission location and the band-gap edge. As the pressure increased in the initial *rhombohedral* phase range, due to the decrease of the distance among the [PbBr₆]⁴⁻ ions upon compression, the electrostatic interaction gradually gained its strength, which decreased the six bond lengths of Pb-Br in same rate. The two Br-Pb-Br angles within [PbBr₆]⁴⁻ octahedra are invariant with pressure (Figure S12, Table S1). Although the bond lengths of Pb-Br decreased with compression, the [PbBr₆]⁴⁻ octahedra exhibited no any distorted configuration below 3.01 GPa and thus no PL response to the external pressure. However, in high-pressure *monoclinic* phase, the three Pb-Br bonds happen to be non-equivalent and the two Br-Pb-Br angles within [PbBr₆]⁴⁻ octahedra persistent change upon compression, demonstrating the distortion of [PbBr₆]⁴⁻ octahedra.

Moreover, as the reviewer referred, DFT calculations in our study show that the Jahn-Teller distortion is around ~0.1 eV, not enough to exhibit emission in the ~520 nm region. However, the origin of broadening emission in our investigation was attributed to the radiative recombination of STEs in mid-band-gap state, rather than an emission from band-gap transition. Therefore, a very large Stokes shift was observed. Overall, there are no direct correlation between the broadening emission and the change of band gaps due to the Jahn-Teller effect.

According to the reviewer's advice, we have made some amendments and added some necessary references in our revised manuscript. The revised details are highlighted in red and can be found in **Line 20-26, Page 8** of the revised manuscript.

REFERENCE

(26) Stoneham, A. M. *et al.* Trapping, self-trapping and the polaron family. *J. Phys.: Condens. Matter* **19**, 255208 (2007).

Comment 6: *The mechanism of PIE under high pressure provided from line 180 is very much unclear and based on fuzzy terminology. The authors attribute the appearance of two peaks at high pressure as the exciton energy and energy gap (!?!). What does this mean??? Why invoking these terms? In bulk structures, you have a band gap, and in molecular like structures you have the optical gap. In NC, you have more than the latter than the former. Especially, for such 0D NC system, which will always exhibit a molecular-like absorption, even in the bulk, because of the disconnection between the octahedra. Meaning the exciton is always confined inside the octahedral (self-trapped). The authors should be more clear in the definition of the terms they introduce. Seeing two peaks and arbitrarily attributing them to exciton and band gap transitions, whatever it means, and even extrapolating the exciton binding energy is quite irritating. As such, also Figure 4 does not make much sense.*

Author reply: We are very sorry for confusing the reviewer due to our unclear statements. In general, the excitonic peak always located a below the band gap absorption, which is usually observed in semiconductor NCs, such as CdSe QDs.^{R1-R3} This will result in a significant difference between the optical absorption edge and the electronic band gap. Only the exciton binding energy is high enough, the excitonic absorption peak can persist at room temperature where normally it would be absent. From the pressure-dependent absorption spectra of Cs₄PbBr₆ NCs, an additional shoulder at the low-energy side of the original absorption band appeared gradually with pressure, consistent with the below-gap absorption (Figure 3). The monoclinic Cs₄PbBr₆ NCs also have the 0D perovskite structure that consists of the disconnected [PbBr₆]⁴⁻ octahedra isolated by Cs⁺ cations, possessing strong quantum confinement of charge carriers. These carriers are always confined inside these octahedra to form bound excitons with high exciton binding energy. Generally, the exciton binding energy can be qualitatively analyzed using the following expression:⁹

$$E_b = \frac{m^* e^4}{2\hbar^2 \epsilon^2}$$

where m^* is the effective mass, e is the electric charge, ϵ is the dielectric constant, and \hbar is Planck constant. We can see that the exciton binding energy is proportional to the value of effective mass of m^* . With the increase in pressure, the skeleton of [PbBr₆]⁴⁻ octahedra gradually compress, which further localizes the charge carrier and increases their effective mass.⁸ Therefore, the exciton binding energy exhibited an increase with the decreased size of octahedra.⁴²⁻⁴⁴ The increase in exciton binding energy ultimately resulted in the appearance of exciton absorption peak at room temperature.

In addition, we calculated the energy band structure and density of states of Cs₄PbBr₆ at 1 atm and 4 GPa by the first-principles calculations, which are performed based on density functional theory implemented in the VASP code. Obviously, the band gap is reduced from 3.23 eV to 3.14 eV, only 0.09 eV,

consistent with the tiny change of the original absorption peak. The theoretical absorption spectrum of Cs_4PbBr_6 was also calculated by the projector-augmented plane-wave (PAW) method as shown in Figure S17. In comparison with the experimental absorption spectra, the theoretical absorption spectrum does not show sharp absorption peaks near the band gap. This suggests that the peak below the band gap absorption cannot be explained simply in terms of the electronic structure of Cs_4PbBr_6 .

In recent reports on low-dimensional perovskites, all of these perovskite materials exhibited a similar broad emission, which were also attributed to the radiative recombination of self-trapped excitons. In addition, the similar excitation absorption peak was also observed and assigned to bound exciton absorption peak in these reports. For instance, Hemamala *et al.* reported that the absorption spectrum of (N-MEDA)[PbBr_4] exhibited a sharp absorption peak at 395 nm with a shoulder at 370 nm, accompanied by a broad emission that spans the entire visible spectrum. The sharp absorption peak was attributed to bound exciton absorption peak in their investigation.¹⁰ Meanwhile, they also reported that the dielectric mismatch between organic and inorganic layers in two-dimensional perovskite (EDBE)[PbX_4] ($X = \text{Cl}, \text{Br}$ and I) leads to strongly bound excitons, reflected in the sharp excitonic absorption bands at 351, 371, and 470 nm for Cl , Br and I , respectively.¹¹ In addition, Mercuri *et al.* also clearly observed excitonic feature in the optical absorption spectra of their two-dimensional perovskites.¹² The exciton binding energy of these low-dimensional perovskite materials exceed 200 meV, consistent with the result deduced from the energy difference between the band gap energy and exciton absorption peak (Figure 3c). Therefore, it is reasonable that the sharp absorption peak at the low-energy side was assigned to exciton absorption band.

In addition, in cases of the strong coupling between the crystal lattice and electrons or holes, a carrier may be self-trapped as a small polaron in its own lattice distortion field. A bound electron-hole pair involving such a carrier is generally described as a self-trapped exciton, and it may dramatically influence luminescence, energy transport, and lattice defect formation in the crystal. Therefore, the self-trapped excitons were generally observed in the deformed crystals. In our investigation, the pressure plays an important role in inducing the distortion of octahedra within Cs_4PbBr_6 NCs. This distortion will locally trap charge carriers, resulting in self-trapped excitons (STEs). The distorted octahedra generate a completely new recombination pathway of excitons via the radiative recombination of STEs.²⁶ Due to the strong quantum confinement of carriers, the excitons were always confined inside octahedra, however, which is different with the self-trapped excitons requiring crystal deformation. The self-trapped state plays an important role to provide a new recombination pathway of excitons.

According to the reviewer's comments, we have made some amendments in the revised manuscript, and modified the schematic illustrations of Figure 4 as configuration coordinate diagram to make it clearer.

The revised details are highlighted in red and can be found in revised Figure 4 of the revised manuscript, and in Line 4-20, Page 21; Line 1-21, Page 22 and Figure S17 of the revised Supporting Information.

REFERENCE

- (R1) Qin, H. *et al.* Single-Dot Spectroscopy of Zinc-Blende CdSe/CdS Core/Shell Nanocrystals: Nonblinking and Correlation with Ensemble Measurements. *J. Am. Chem. Soc.* **136**, 179-187, (2014).
- (R2) Washington, A. L. *et al.* Ostwald's Rule of Stages and Its Role in CdSe Quantum Dot Crystallization. *J. Am. Chem. Soc.* **134**, 17046-17052, (2012).
- (R3) Subila, K. B., Kishore Kumar, G., Shivaprasad, S. M. & George Thomas, K. Luminescence Properties of CdSe Quantum Dots: Role of Crystal Structure and Surface Composition. *J. Phys. Chem. Lett.* **4**, 2774-2779, (2013).
- (8) L., M. R., E., E. G., J., S. H., B., J. M. & M., H. L. Temperature - Dependent Charge - Carrier Dynamics in CH₃NH₃PbI₃ Perovskite Thin Films. *Adv. Funct. Mater.* **25**, 6218-6227, (2015).
- (9) Miyata, A. *et al.* Direct measurement of the exciton binding energy and effective masses for charge carriers in organic-inorganic tri-halide perovskites. *Nat. Phys.* **11**, 582, (2015).
- (10) Dohner, E. R., Jaffe, A., Bradshaw, L. R. & Karunadasa, H. I. Intrinsic White-Light Emission from Layered Hybrid Perovskites. *J. Am. Chem. Soc.* **136**, 13154-13157, (2014).
- (11) Dohner, E. R., Hoke, E. T. & Karunadasa, H. I. Self-Assembly of Broadband White-Light Emitters. *J. Am. Chem. Soc.* **136**, 1718-1721, (2014).
- (12) Mao, L., Wu, Y., Stoumpos, C. C., Wasielewski, M. R. & Kanatzidis, M. G. White-Light Emission and Structural Distortion in New Corrugated Two-Dimensional Lead Bromide Perovskites. *J. Am. Chem. Soc.* **139**, 5210-5215, (2017).
- (26) Stoneham, A. M. *et al.* Trapping, self-trapping and the polaron family. *J. Phys.: Condens. Matter* **19**, 255208 (2007).
- (42) Ramvall, P., Tanaka, S., Nomura, S., Riblet, P. & Aoyagi, Y. Observation of confinement-dependent exciton binding energy of GaN quantum dots. *Appl. Phys. Lett.* **73**, 1104-1106, (1998).
- (43) Meulenbergh, R. W. *et al.* Determination of the Exciton Binding Energy in CdSe Quantum Dots. *ACS Nano* **3**, 325-330, (2009).
- (44) Takagahara, T. & Takeda, K. Theory of the quantum confinement effect on excitons in quantum dots of indirect-gap materials. *Phys. Rev. B* **46**, 15578-15581, (1992).

Figure 3. (a) Typical profile of absorption band against pressures for Cs₄PbBr₆ NCs in situ measured in a DAC apparatus. The dashed arrows indicates the shift of absorption bands of E_{ex} and E_g. (b) Absorption spectra of Cs₄PbBr₆ NCs measured at a selected pressure of 6.17 GPa. Therein, E_{ex}, E_g, and E_b represent the exciton energy, the energy gap, and the exciton binding energy, respectively. (c) Pressure dependence of exciton binding energy E_b deduced from the difference of E_{ex} and E_g.

Added Figure S17. The absorption spectra of Cs₄PbBr₆ at 6.17 GPa along with the theoretical absorption-coefficient spectrum calculated by the PAW method.

Revised Figure 4. Configuration coordinate model of PL enhancement for the high-pressure monoclinic phase of Cs_4PbBr_6 NCs at 4 GPa (a) and 6 GPa (b). Therein, the absorption transition upon excitation is described from A to B. The STEs recombination emission is depicted from C to D. Two distinct paths at 4 GPa are represented: thermal overcoming of the barrier (1) and tunneling (2). (FC): free carrier state, (BE): bound exciton state, ΔQ : the equilibrium position difference between ground and self-trapped exciton state.

Comment 7: *The appearance of two peaks can be attributed to for example, two types of phases with different absorption onsets. The XRD spectra in the supporting info also indicate that this phase is quite amorphous, meaning that broadening and splitting of peaks are very likely. Another explanation to the emission is that under pressure, you may distort so much the lattice that you could lead to connection between octahedra, and the formation of impurities of 3D structures. You cannot discard this for two reasons: the red shift in absorption and the amorphous structure you find at these high pressures.*

Author reply: Thanks very much for the reviewer's insightful comments and we are sorry for confusing the reviewer. Firstly, the reviewer doubted that the appearance of two peaks were attributed to two types of phases with different absorption onsets, because the broadening and splitting of peaks are very likely due to the amorphous feature of high-pressure phase. However, from the in situ high pressure synchrotron ADXRD patterns of the Cs_4PbBr_6 NCs (Figure S4), we can see that a pure high-pressure monoclinic structure was completely achieved at 4.01 GPa, at which the ADXRD pattern can be well fitted by Rietveld refinements ($R_{wp}=1.76\%$, $R_p=1.36\%$). Furthermore, obvious diffraction rings can be also observed at 4.01 and 6.12 GPa, respectively, indicating that the high-pressure phase in this study was well crystalline, not amorphous. The broadening of diffraction peaks at high-pressure phase should be highly related to the deviatoric stress which would inevitably produce the broad diffraction peaks under higher

pressures.^{R1-R3} Furthermore, the amorphous perovskites generally exhibited no fluorescence feature,^{R4-R6} which also disagreed with the emission enhancement at high-pressure condition. Therefore, the possibility that “*The appearance of two peaks can be attributed to two types of phases with different absorption onsets due to the amorphous structure*” could be excluded.

Secondly, the reviewer doubted that the lattice may be distorted so much to lead to the connection between the octahedra, and thus the formation of impurities of 3D perovskite structures under pressure. However, the recent studies on 3D CsPbBr₃ with compression demonstrated that the PL intensity of CsPbBr₃ was generally decreased with pressure and completely disappeared above ~2 GPa.^{R4,R7} Whereas in this work, the emission of the compressed Cs₄PbBr₆ NCs appears suddenly at 3.01 GPa, which exceeds the pressure point of fluorescence quenching in CsPbBr₃. In addition, the subsequent, PIEE of 0D Cs₄PbBr₆ NCs was also opposite to the PL response of 3D CsPbBr₃ under high pressure. Furthermore, both the CsPbBr₃ bulk and nanomaterials underwent a redshift below ~1.2 GPa, and then a blueshift until the vanishing of PL intensity with further compression. The evolution of absorption spectra for 3D CsPbBr₃ NCs was also opposite to the change of the new absorption peak for Cs₄PbBr₆ NCs at high pressure. Finally, for 3D CsPbBr₃, the FWHM of emission was always less than 30 nm, whereas the FWHM of present Cs₄PbBr₆ NCs was even up to 210 nm, a very broad emission. Accordingly, the formation of 3D structure in Cs₄PbBr₆ NCs at high pressure can be reasonably ruled out from the above analysis.

Revised Figure S4. (a) Representative in situ high-pressure XRD patterns of Cs₄PbBr₆ NCs with the presence of silicon oil as PTM during the high-pressure experiments. (b) Rietveld refinements of the experimental (red fork), simulated (black profile), and difference (black line) XRD patterns of phase I at ambient conditions and phase II at the pressure of 4.01 and 6.12 GPa. Therein, blue vertical markers indicate the corresponding Bragg reflections.

REFERENCE

- (R1) Wang, T. *et al.* Pressure Processing of Nanocube Assemblies Toward Harvesting of a Metastable PbS Phase. *Adv. Mater.* **27**, 4544-4549, (2015).
- (R2) Zhu, J. *et al.* Structural evolution and mechanical behaviour of Pt nanoparticle superlattices at high pressure. *Nanoscale* **8**, 5214-5218, (2016).
- (R3) Wang, Z. *et al.* Deviatoric Stress Driven Formation of Large Single-Crystal PbS Nanosheet from Nanoparticles and in Situ Monitoring of Oriented Attachment. *J. Am. Chem. Soc.* **133**, 14484-14487, (2011).
- (R4) Xiao, G. *et al.* Pressure Effects on Structure and Optical Properties in Cesium Lead Bromide Perovskite Nanocrystals. *J. Am. Chem. Soc.* **139**, 10087-10094, (2017).
- (R5) Wang, Y.; Lü, X.; Yang, W.; Wen, T.; Yang, L.; Ren, X.; Wang, L.; Lin, Z.; Zhao, Y. *J. Am. Chem. Soc.* **2015**, *137*, 11144.
- (R6) Kong, L. *et al.* Simultaneous band-gap narrowing and carrier-lifetime prolongation of organic–inorganic trihalide perovskites. *Proc. Natl. Acad. Sci. U. S. A.* **113**, 8910-8915, (2016).
- (R7) Zhang, L., Zeng, Q.; Wang, K. Pressure-Induced Structural and Optical Properties of Inorganic Halide Perovskite CsPbBr₃. *J. Phys. Chem. Lett.* **8**, 3752-3758, 7b01577 (2017).

In summary, following the kind suggestions and insightful comments of the reviewer, we carefully rechecked our manuscript and made some amendments to response to the reviewer's issues. Hopefully we have addressed all of your concerns. The corresponding revised details are highlighted in red and can be found in Line 20-29, Page 5; Line 3-5, Page 6; Line 16-30, Page 7; Line 1-26, Page 8 and Figure 4 of the revised manuscript; Line 4-16, Page 12; Line 1-6, Page 13; Line 4-20, Page 21; Line 1-21, Page 22; Figure S4; Figure S10; Figure S11; Figure S17; Table S1 and their captions in the revised Supporting Information.

Reviewers' Comments:

Reviewer #1:

Remarks to the Author:

The revision well addressed my concerns. With such significant improvement, I would like to recommend it with an acceptance for publication at Nature Communications.

Reviewer #2:

Remarks to the Author:

The issues pointed out earlier have been responded satisfactorily. The revised manuscript is much improved and recommended for publication.

Reviewer #3:

Remarks to the Author:

I appreciate the detailed answers from the authors of the paper, and again I restate my interest for these experiments. However I say with regret that in the current form this article is not suitable for publication in Nature Communications.

This work still presents major flaws in the interpretation of the measured data that in many cases are simply wrong. The answers provided to my previous report, which I appreciate, are not satisfactory. In addition, much of the terminology used is again not clear and hard to decode.

The major flaw I find in this work is in the interpretation of the absorption spectra of Figure 3 obtained after exerting pressure on the NCs. Again, the authors keep separating band-edge energy absorption and excitonic absorption peak. Let me state this clear, this interpretation is plain wrong. OD structures present ONLY strongly bound excitons, which is exemplified by the clear and sharp excitonic peak in the absorption spectrum of this material without pressure (Figure 3, bottom). The band-edge absorption, typical of "free-carrier" like absorption, does not exist for this material. And there is no way that you can separate between the two if the material is deformed upon pressure.

The most reasonable explanation for the formation of these two peaks is that upon distorting the octahedron, you are breaking the symmetry of the [PbBr₆] unit already in the ground state. This is highlighted by the calculations presented in Table S1. Here you form two classes of Pb-Br bonds, which very likely lead to the splitting of the excitonic peak in two: one at higher and one at lower energies, easily understood from the different bond lengths you compute.

As a consequence, the appearance of the emission peak at high pressure can be attributed to the fact that, being the octahedron already distorted, the Stokes shift (or reorganization due to photoinduced Jahn Teller effect, or polaronic effect, they are all synonymous) is smaller than the "unpressured" OD structure. This allows a reasonable wavefunction overlap between the excited structure and the ground state one, leading to a likely non-zero or larger transition dipole moment (oscillator strength) compared to the ideal OD structure. By the way, I also requested in the previous report if the authors could provide a computation of the transition dipole moments, but failed to do so.

Another major flaw is in the misuse of the word STE, self-trapped emission. The author state that this is an emission from a mid-band-gap state. This is also wrong. A mid-gap state is a localized state that is usually formed by defects in the material or in this case by a defect on the octahedron (e.g. Br vacancy). A simple reorganization of the excited state structure of this material upon photo-excitation, even if large as in this case, does not represent an emission from a midgap

state, but it is simply an emission from a state with a large Stokes shift (due to photoinduced Jahn-Teller effect, etc...). For this reason, the answer given in the rebuttal to Comment 5 (last paragraph) is full contradictory: "Jahn-Teller distortion is around 0.1 eV, , however the origin of the broadening is due ... STE in mid-band gap state rather than emission in band-edge. Therefore, a very large Stokes shift is observed".

Photinduced Jahn-Teller distortion, STE emission in mid-gap (as the authors defines) and Stokes shift are all the same thing!!! The only difference is that when pressure is induced some distortion occurs in the ground-state (not necessary due to Jahn-Teller).

Referee: #1

Comments: The revision well addressed my concerns. With such significant improvement, I would like to recommend it with an acceptance for publication at Nature Communications.

Author reply: We are pleased that we have sufficiently addressed all the concerns of the referee. We thank again the referee for his/her valuable comments that greatly improve the quality of the manuscript.

Referee: #2

Comments: The issues pointed out earlier have been responded satisfactorily. The revised manuscript is much improved and recommended for publication.

Author reply: We are delighted that all the issues pointed out by the referee have been satisfactorily addressed. We thank the referee again for his/her useful and constructive comments.

Reviewer: #3

Comment 1: I appreciate the detailed answers from the authors of the paper, and again I restate my interest for these experiments. However I say with regret that in the current form this article is not suitable for publication in Nature Communications. This work still presents major flaws in the interpretation of the measured data that in many cases are simply wrong. The answers provided to my previous report, which I appreciate, are not satisfactory. In addition, much of the terminology used is again not clear and hard to decode.

Author reply: We thank the referee for his/her consistent interest in our work and for sharply pointing out the issues of the manuscript. In response to the referee's clear and constructive comments, we have revised extensively the contents on the interpretation of the measured data. Additional calculations were also performed to reveal the underlying physical mechanism. With these revisions, we feel that we have achieved a more thorough understanding of the mechanism responsible for our experimental observation. We highly appreciate the opportunity the referee offered to further improve the quality of the manuscript.

Comment 2: The major flaw I find in this work is in the interpretation of the absorption spectra of Figure 3 obtained after exerting pressure on the NCs. Again, the authors keep separating band-edge energy absorption and excitonic absorption peak. Let me state this clear, this interpretation is plain wrong. 0D structures present ONLY strongly bound excitons, which is exemplified by the clear and sharp excitonic peak in the absorption spectrum of this material without pressure (Figure 3, bottom). The band-edge absorption, typical of "free-carrier" like absorption, does not exist for this material. And there is no way that you can separate between the two if the material is deformed upon pressure.

The most reasonable explanation for the formation of these two peaks is that upon distorting the octahedron, you are breaking the symmetry of the [PbBr₆] unit already in the ground state. This is highlighted by the calculations presented in Table S1. Here you form two classes of Pb-Br bonds, which very likely lead to the splitting of the excitonic peak in two: one at higher and one at lower energies, easily understood from the different bond lengths you compute.

Author reply: We are grateful to the referee for pointing out this issue and offering constructive suggestion on revision. Thanks to the referee's comment, we did recognize that the 0D Pb halide (X) perovskites cannot have excitonic peak emerging from band-edge absorption, but present only strongly

bound excitons. This is because the “free-carrier” like band-edge absorption would never exist in the 0D system with isolated molecular-like PbX_6 octahedra units providing strong confinement to photoexcited carriers. Inspired by the referee’s insightful comment, we have now assigned two absorption peaks at high pressures in Figure 3 to the two splitted excitonic peaks. This assignment is indeed reasonable by considering the broken symmetry of the distorted $[\text{PbBr}_6]^{4-}$ units in the high-pressure ground-state structure.

Moreover, we obtained further evidence by the newly calculated absorption oscillator strengthes using the excited-state structure associated with self-trapped exciton (Figure S14) at the single-particle level: as anticipated by the referee, in the lower energy region the high-pressure phase shows two separated large oscillator strengthes, implying the signature of peak splitting, while the ambient-pressure phase shows one group of oscillator strengthes. Therefore the symmetry breaking of the $[\text{PbBr}_6]^{4-}$ units at high pressure may be indeed responsible for the emergence of two excitonic peaks in absorption spectrum.

In the revised manuscript, we have modified Figure 3, added Figure S14 into Supporting Information, and made the following changes of discussions at Line 5-29, Page 13 in manuscript:

We attributed the formation of two absorption peaks to the splitting of the excitonic peak. As the absorption bands of perovskites are greatly influenced by the nature of the octahedra. Therefore, the formation of these two absorption peaks should be highly related to the distortion of $[\text{PbBr}_6]^{4-}$ octahedra. As shown in Table S1, although the bond lengths of Pb-Br decreased with compression, the $[\text{PbBr}_6]^{4-}$ octahedra exhibited no any distorted configuration below 3.01 GPa. However, in high-pressure monoclinic phase, the three Pb-Br bonds happen to be non-equivalent within $[\text{PbBr}_6]^{4-}$ octahedra persistent change upon compression, forming two classes of Pb-Br bonds. Therein, the breaking symmetry of the $[\text{PbBr}_6]^{4-}$ unit in the ground state under pressure was very likely lead to the splitting of the excitonic peak in two: one at higher and one at lower energies. This assignment is further supported by the absorption oscillator strengthes using the excited-state structure associated with self-trapped exciton (Figure S14) at the single-particle level. We found that in the lower energy region the high-pressure phase shows two separated large oscillator strengthes, implying the signature of peak splitting, while the ambient-pressure phase shows one group of oscillator strengthes. Likewise, the energy difference ΔE between the two exciton absorption peaks (Figure 3b) can also indicate the extent of distortion for octahedra with increasing pressure. As illustrated in Figure 3c, we find that this energy difference sharply increases in the pressure region from 4.03 to 6.17 GPa with a pressure coefficient of 0.08 eV/GPa (Figure 3c). When the pressure increases beyond 6.17 GPa, the ΔE experienced a relatively sluggish increase. Figure 3c demonstrated that the distorted extent of octahedra was

improved with compression, but beyond 6.17 GPa, the $[\text{PbBr}_6]^{4-}$ octahedra started to tilt and undergo a considerable rotation deviating from the original orientation in disordered way, consistent with the result of high-pressure angle dispersive synchrotron X-ray diffraction.

Revised Figure 3. (a) Typical profile of absorption band against pressures for Cs_4PbBr_6 NCs in situ measured in a DAC apparatus. The dashed arrows indicate the shift of bound exciton absorption peaks of E_{ex1} and E_{ex2} . (b) Absorption spectra of Cs_4PbBr_6 NCs measured at a selected pressure of 6.17 GPa. Therein, E_{ex1} , and E_{ex2} represent the splitting bound exciton absorption peaks. (c) Pressure dependence of energy difference ΔE deduced from the difference of E_{ex1} and E_{ex2} .

Added Figure S14. Calculated absorption oscillator strengths using the excited-state structure associated with self-trapped exciton at 1 atm (a) and 4 GPa (b), respectively. The oscillator strengths, *i.e.* transition dipole moments versus photon energy is shown. We note that upon the appearance of the self-trapped exciton, the excited and ground states are many-body wavefunctions involving electron-hole pairs and lattice distortion, therefore the strictly accurate calculation of transition dipole moments between them is complicated and challenging, which is beyond the scope of current study. For simplicity we took the excited-state structure associated with self-trapped exciton, *i.e.*, the lowest-energy spin-triplet state in company with lattice distortion, and calculate the transition dipole moments at the single-particle level. The calculated results, though approximated, are expected to capture the main feature of the realistic light emission process of self-trapped exciton. It should be also noted that the photon energies in the plots are not accurate in the single-particle approximation.

Comment 3: *As a consequence, the appearance of the emission peak at high pressure can be attributed to the fact that, being the octahedron already distorted, the Stokes shift (or reorganization due to photoinduced Jahn Teller effect, or polaronic effect, they are all synonymous) is smaller than the “unpressured” 0D structure. This allows a reasonable wavefunction overlap between the excited structure and the ground state one, leading to a likely non-zero or larger transition dipole moment (oscillator strength) compared to the ideal 0D structure. By the way, I also requested in the previous report if the authors could provide a computation of the transition dipole moments, but failed to do so.*

Author reply: We thank the referee for making this good suggestion. As the referee pointed out, since high pressure causes the distortion of the $[\text{PbBr}_6]^{4-}$ octahedron, the structure relaxation required to trap a photoexcited exciton is expected to be smaller than that in the ambient-pressure phase containing perfect $[\text{PbBr}_6]^{4-}$ octahedra. The resulted smaller Stokes shift may indeed allow reasonable wavefunction overlap

between the excited state (associated with self-trapped exciton) and the ground state, leading to a likely non-zero or enhanced transition dipole moment compared to the ambient-pressure phase. Following the referee's this suggestion and also the request in the previous report, we calculated the transition dipole moments based on the excited-state structure associated with self-trapped exciton, as shown in the above Figure S14. We note that upon the appearance of the self-trapped exciton, the excited and ground states are many-body wavefunctions involving electron-hole pairs and lattice distortion, therefore the strictly accurate calculation of transition dipole moments between them is complicated and challenging, which is beyond the scope of current study. For simplicity we took the excited-state structure associated with self-trapped exciton, *i.e.*, the lowest-energy spin-triplet state in company with lattice distortion, and calculate the transition dipole moments at the single-particle level. The calculated results, though approximated, are expected to capture the main feature of the realistic light emission process of self-trapped exciton. We found that the high-pressure phase exhibits a one-magnitude larger oscillator strength (~ 0.06 a.u.) at the lowest excitation energy than that of the ambient-pressure phase (~ 0.003 a.u.) This implies that the pressure-induced distortion of the $[\text{PbBr}_6]^{4-}$ octahedra promotes the self-trapping excitonic state to be more optically active.

In addition to some extent enhanced optical activity, our further first-principle calculations show that the self-trapped exciton in the high-pressure phase has the larger exciton binding energy (1.31 eV) than that of the ambient-pressure phase (1.13 eV). This originates from the increased electron-phonon coupling strength when the $[\text{PbBr}_6]^{4-}$ octahedra are contracted upon compression, accompanying with the strengthened Pb-Br covalent bondings (see Table S1). Since the self-trapped exciton is mediated by the interaction between exciton and lattice distortion, the stronger electron-phonon coupling can more effectively bind photoexcited carriers to form self-trapped exciton in the lattice distortion field. The revised Figure S15 shows the comparison of electronic band structures with and without the lattice distortion mediating self-trapped exciton for the ambient-pressure (1 atm) and high-pressure (4 GPa), respectively. As seen, the electronic band structure is much more seriously distorted upon the presence of lattice distortion for the high-pressure phase, which ambiguously indicates the stronger electron-phonon coupling. Such increased electron-phonon coupling strength is responsible for the larger binding energy of self-trapped exciton in the high-pressure phase.

Accordingly, we have added the following discussion into **Line 30, 13** and **Line 1-28, Page 14**:

Moreover, since high pressure causes the distortion of the $[\text{PbBr}_6]^{4-}$ octahedron, the excited state structural reorganization required to trap a photoexcited exciton is expected to be smaller than that in the

ambient-pressure phase containing perfect $[\text{PbBr}_6]^{4-}$ octahedra. The resulted smaller Stokes shift may allow reasonable wavefunction overlap between the excited state (associated with self-trapped exciton) and the ground state, leading to a likely non-zero or enhanced transition dipole moment compared to the ambient-pressure phase. Given that, we calculated the transition dipole moments based on the excited-state structure associated with self-trapped exciton at the single-particle level, as shown in Figure S14. We found that the high-pressure phase exhibits a one-magnitude larger oscillator strength (~ 0.06 a.u.) at the lowest excitation energy than that of the ambient-pressure phase (~ 0.003 a.u.) This implies that the pressure-induced distortion of the $[\text{PbBr}_6]^{4-}$ octahedra promotes the self-trapping excitonic state to be more optically active.

In addition to some extent enhanced optical activity, our further first-principle calculations show that the self-trapped exciton in the high-pressure phase has the larger exciton binding energy (1.31 eV) than that of the ambient-pressure phase (1.13 eV). This originates from the increased electron-phonon coupling strength when the $[\text{PbBr}_6]^{4-}$ octahedra are contracted upon compression, accompanying with the strengthened Pb-Br covalent bondings (see Table S1). Since the self-trapped exciton is mediated by the interaction between exciton and lattice distortion, the stronger electron-phonon coupling can more effectively bind photoexcited carriers to form self-trapped exciton in the lattice distortion field. The Figure S15 shows the comparison of electronic band structures with and without the lattice distortion mediating self-trapped exciton for the ambient-pressure (1 atm) and high-pressure (4 GPa), respectively. As seen, the electronic band structure is much more seriously distorted upon the presence of lattice distortion for the high-pressure phase, which ambiguously indicates the stronger electron-phonon coupling. Such increased electron-phonon coupling strength is responsible for the larger binding energy of self-trapped exciton in the high-pressure phase.

Revised Figure S15. Electronic band structure of Cs_4PbBr_6 with and without the lattice distortion mediating self-trapped exciton for the ambient-pressure (1 atm) and high-pressure (4 GPa) phases, respectively. The band structure of equilibrium structure (black lines) shifts upon the presence of lattice distortion (red lines). Note that the lattice distortion magnitude adopted in the calculations are the same for the ambient-pressure and high-pressure phases.

Comment 4: Another major flaw is in the misuse of the word STE, self-trapped emission. The author state that this is an emission from a mid-band-gap state. This is also wrong. A mid-gap state is a localized state that is usually formed by defects in the material or in this case by a defect on the octahedron (e.g. Br vacancy). A simple reorganization of the excited state structure of this material upon photo-excitation, even if large as in this case, does not represent an emission from a midgap state, but it is simply an emission from a state with a large Stokes shift (due to photoinduced Jahn-Teller effect, etc...). For this reason, the answer given in the rebuttal to Comment 5 (last paragraph) is full contradictory: “Jahn-Teller distortion is around 0.1 eV, , however the origin of the broadening is due ... STE in mid-band gap state rather than emission in band-edge. Therefore, a very large Stokes shift is observed”. Photoinduced Jahn-Teller distortion, STE emission in mid-gap (as the authors defines) and Stokes shift are all the same thing!!! The only difference is that when pressure is induced some distortion occurs in the ground-state (not necessary due to Jahn-Teller).

Author reply: We appreciate the referee for kindly pointing out the issue of our using impertinent and misleading terminologies to explain experimental observations. In response we have carefully revised the manuscript by eliminating the impertinent terminologies mentioned by the referee. Briefly we made the following changes:

(i) We fully agree with the referee that the terminologies of “self-trapped exciton”, “photoinduced Jahn-Teller distortion”, “a state with a large Stokes shift”, and “excited state structural reorganization” mean the same content in physics. We have now given a clear definition of “self-trapped exciton” by utilizing these terminologies in **Line 20-22, Page 8 and Line 14-20, Page 9**: Self-trapped exciton (STE) represents a photoinduced electron-hole pair (exciton) mediated by the interaction between exciton and corresponding lattice. In addition, in this 0D perovskite materials, the Cs⁺ cations isolate each [PbBr₆]⁴⁻ octahedra. Thus, the Cs₄PbBr₆ is considered to exhibit the intrinsic properties of the individual [PbBr₆]⁴⁻ octahedra, and the photoluminescent properties are explained to be not as a result of lattice defects but rather due to excited state structural reorganization within individual [PbBr₆]⁴⁻ octahedra. (Manna *et al.*, *Nano Lett.* **2017**, *17*, 1924; Manna *et al.*, *J. Phys. Chem. Lett.* **2018**, *9*, 2326.) Such excited state structural reorganization in this 0D perovskites is related with the exciton self-trapping. (Ma *et al.*, *Chem. Sci.* **2018**, *9*, 586.) Because of the photoinduced Jahn-Teller distortion, which causes substantial octahedral distortion upon photoexcitation, the STE usually acts as a state a large Stokes shift.

(ii) We then use the well-defined terminology of self-trapped exciton (STE) to explain our experimental observations throughout the manuscript.

(iii) The misleading terminology of “mid-band-gap state” has been removed.

(iv) The effect of pressure is causing the distortion of the [PbBr₆]⁴⁻ octahedron, which (i) facilitates the photoinduced Jahn-Teller effect, which causes the smaller Stokes shift and thus enhanced optical activity of STE compared with the ambient-pressure phase, and (ii) causes the increased electron-phonon coupling strength and hence the larger exciton binding energy. Based on our analyses these two facts are responsible for the appearance of the exciton emission peak at high pressure.

(v) To avoid confusion, we determined to not use the terminology of “the Jahn-Teller effect” to depict the structure distortion of the ground-state structure before photoexcitation, which is simply caused by pressure-induced lattice contraction and atoms rearrangement.

**Pressure-Induced Emission of Halide Perovskite Cs₄PbBr₆ Nanocrystals**

Zhiwei Ma,¹ Zhun Liu,² Siyu Lu,³ Lingrui Wang,¹ Xiaolei Feng,⁴ Dongwen Yang,² Kai Wang,¹
Guanjun Xiao,^{1,*} Lijun Zhang,^{2,*} Simon A. T. Redfern,⁴ and Bo Zou^{1,*}

¹*State Key Laboratory of Superhard Materials, College of Physics, Jilin University, Changchun*
*130012, China*

²*Key Laboratory of Automobile Materials of MOE, and College of Materials Science, Jilin*
*University, Changchun 130012, China*

³*College of Chemistry and Molecular Engineering, Zhengzhou University Zhengzhou 450001,*
*China*

⁴*Department of Earth Sciences, Downing Street, University of Cambridge, Cambridge, CB2 3EQ,*
*UK*

*Corresponding author. Email: xguanjun@jlu.edu.cn (G.X.); lijun_zhang@jlu.edu.cn (L.Z);
zoubo@jlu.edu.cn (B.Z.)

**ABSTRACT:** Metal halide perovskites (MHPs) are of great interest because of their
high conversion efficiency in optoelectronic devices and solar cells. However,
exploring an effective strategy to further improve their optical properties remains a
considerable challenge. In this regard, the initially non-fluorescent zero-dimensional
(0D) Cs₄PbBr₆ nanocrystals (NCs) happen to exhibit a distinct emission at 3.01 GPa.
Subsequently, the emission intensity of Cs₄PbBr₆ NCs experiences a significant
increase upon further compression. Such pressure-induced emission (PIE) is attributed
to radiative recombination of the self-trapped excitons, which are emerged associated
with the large distortion of Pb-Br octahedra resulting from a structural phase
transition. **Joint experimental and theoretical analyses indicate that the emergence of**
**room temperature luminescence may be ascribed to the enhanced optical activity and**
**the increased binding energy of self-trapping excitons upon compression.** Our
findings render high pressure as a robust tool to boost the photoluminescence
efficiency and provide insights into the relationship between the structure and optical
properties of 0D MHPs at extremes.

**1. Introduction**

The recent success of organometallic halide perovskite nanocrystals (NCs) in
photovoltaic devices has further triggered research activities on inorganic metal halide
perovskites (MHPs) NCs due to the good stability than their organic counterparts.¹⁻⁶
Cs_4PbBr_6 is a typical zero-dimensional (0D) inorganic MHP, in which the octahedra
are completely isolated by cation bridges and charge carriers are localized within the
ordered metal halide component.⁷⁻¹⁰ The electronic structure and bonding patterns of
these materials are expected to show significant dependence on pressure.¹¹⁻¹⁶ In view
of this, we explored their pressure dependence and find, for example, that the band
gap alignment of CsPbBr_3 NCs can be successfully fine-tuned by means of pressure.¹⁷
In addition, Chen et al. have demonstrated that pressure-sintered CsPbBr_3
nanoplatelets show a 1.6-fold enhancement in photoluminescence (PL) and display
longer emission lifetimes than the untreated NCs.¹³ However, such studies have been
largely limited to 3D perovskites, without considerations of any
low-dimensional-network analogues. Furthermore, developing an effective strategy to
improve the optical properties of MHPs remains a pressing challenge.

To our knowledge, the pressure-dependent behavior of 0D perovskites has not
previously been investigated, although it may have significant effects. We have,
therefore, carried out systematic high-pressure studies of a typical 0D perovskite,
Cs_4PbBr_6 NCs. We find that the Cs_4PbBr_6 NCs undergo a structural phase transition
from rhombohedral to monoclinic structure upon compression. Energetic calculations
indicate that the monoclinic structure is energetically more favorable than the
rhombohedral phase with increasing pressure. The $[\text{PbBr}_6]^{4-}$ octahedra are
significantly distorted after phase transition, which provides a condition for emerging
novel optical properties. As a result, the Cs_4PbBr_6 NCs exhibit an unexpected
pressure-induced emission (PIE) at room temperature when the originally intrinsic
non-emitting material is compressed to 3.01 GPa. The underlying mechanism is
attributed to the formation of the self-trapped excitons, which are emerged associated
with the large distortion of Pb-Br octahedra upon compression. **With the aid of**
**first-principle calculations, we found the enhanced optical activity and the increased**

binding energy of self-trapping excitonic states may be responsible for the room
 temperature luminescence. Our results suggest that pressure processing offers an
 exciting means to access new types of perovskite materials which may show enhanced
 functional properties by overcoming the limitations of conventional synthetic
 chemistry.

2. Results and Discussion

2.1 The morphology and structure of Cs_4PbBr_6 NCs at ambient condition

**Figure 1.** (a) TEM image and (b) high-resolution TEM image of Cs_4PbBr_6 NCs before
 compression. (c) HAADF-STEM image and element mapping (Cs, Pb and Br) of the as-prepared
 Cs_4PbBr_6 NCs. Schematic crystal structure of Cs_4PbBr_6 along (e) and perpendicular (f) to [001].

As shown in Figure 1a and b, the samples before compression exhibit a

well-defined morphology with good monodispersity. It is found that the Cs_4PbBr_6
NCs have an average diameter of 14.4 nm with a standard deviation of 1.3 nm. The
measured lattice spacing of 0.68 nm from HRTEM image (Figure 1b) corresponds to
the spacing of the (110) planes of rhombohedral Cs_4PbBr_6 . Elemental mapping
indicates that the elements of Cs, Pb and Br are homogeneously distributed
throughout the whole sample, which suggests high purity of the final products (Figure
1c). Figure 1d and e illustrate the crystal structures of 0D perovskite Cs_4PbBr_6 in its
rhombohedral phase along and perpendicular to the c axis. Within the Cs_4PbBr_6 NCs,
the $[\text{PbBr}_6]^{4-}$ octahedra are completely decoupled in all dimensions, as a result of
minimal electronic overlap between the adjacent octahedra. Therefore, 0D perovskite
NC system always exhibit a molecular-like absorption, and the size of the NCs does
not have any remarkable effect on the band structure, which is well consistent with
the report by Manna.^{8,18} Meanwhile, this unique structure renders a strong quantum
confinement to confine charge carriers inside the octahedra to easily form bound
excitons, which is exemplified by the clear and sharp excitonic peak in the absorption
spectrum of this material without pressure, as shown in Figure S1. We further
calculated the band structure and density of states of ambient-pressure Cs_4PbBr_6 by
the first-principles calculations (Figure S2). The separation of the conduction band
minimum (CBM) and the valence band maximum (VBM) at the Z point of the
Brillouin zone gives a direct band gap of ~ 3.23 eV. Note that spin-orbit coupling in
our calculations significantly reduces the band gap by inducing a large splitting of the
first degenerated conduction levels. The calculated total and partial density of states
demonstrates that the valence bands primarily originate from the $4p$ orbitals of Br,
while the conduction band is mainly contributed by the $6p$ orbitals of Pb. In addition,
the steady-state PL of samples was performed at ambient condition and no emission
was observed, matching with previous reports on bulk Cs_4PbBr_6 powders and
films.^{19,20} The absence of luminescence at room temperature of Cs_4PbBr_6 has been
attributed to the thermal quenching effect^{8,19} that is caused by exciton migration²¹ or
nonradiative recombination process involving phonons¹⁹, etc. Although some works
showed that Cs_4PbBr_6 NCs or the bulk structure possessed fluorescence in the visible

region,²²⁻²⁴ there remains ongoing debate about the origin of the emission for these
systems. The recent report by Manna *et al.*¹⁸ proposed a more reasonable explanation,
deepening insight into the non-fluorescent mechanism of the Cs₄PbBr₆ systems.
According to their reports,^{8,18} the origin of PL can be attributed to the presence of 3D
perovskite impurities embedded inside the Cs₄PbBr₆ matrix, rather than the intrinsic
emission of Cs₄PbBr₆.

**2.2 In situ high-pressure photoluminescence measurement of Cs_4PbBr_6 NCs**

 **Figure 2.** (a), (b) and (c) Changes in the PL spectra of Cs_4PbBr_6 NCs under pressure. Black arrows
 indicate the evolution of the PL spectra as a function of pressure. (d) Pressure-dependent
 chromaticity coordinates of the emissions

 It has been suggest previously that the PL emission of perovskites might be
 greatly influenced by the nature of the octahedra.²⁵ Accordingly, intriguing PL
 properties might be expected as a result of structural modulation under varying

pressure. For our sample, the pressure-dependent PL spectra were recorded up to
18.23 GPa (Figure 2). We can observe that Cs₄PbBr₆ NCs initially exhibit no PL
response to the external pressure below 3.01 GPa, after which a broad emission band
with a full width at half maximum (FWHM) of ~150 nm appears suddenly (Figure 2a).
This drastic change should be governed by the structure-related transformation. On
further compression, the fluorescence of Cs₄PbBr₆ NCs exhibits an unambiguous
pressure-sensitive evolution. Note that the corresponding PL intensity shows a
remarkable increase with increasing pressure, eventually reaching a maximum at the
pressure of 6.23 GPa (Figure 2b, Figure S3a). Pressure dependence of the PL
wavelength and the FWHM in our Cs₄PbBr₆ NCs is shown in Figure S3b and S3c. We
can see that the PL peak of Cs₄PbBr₆ NCs shows an initial blue-shift upon
compression to 6.23 GPa. Thereafter, it displays a redshift to 18.23 GPa, at which
point the PL intensity almost disappears (Figure 2c). The FWHM of the PL peak
sharply decreases in the pressure region from 3.01 to 6.17 GPa, followed by a
relatively sluggish increase.

The profile of the emission under high-pressure conditions appears asymmetric
and skewed to the low-energy side with a very large Stokes shift exceeding 190 nm.
The recombination between self-trapped excitons (STEs) is a well known mechanism
accounting for large Stokes shifts for a number of broad emitting lead halide
perovskites, such as the 2D, 1D and 0D system in recent reports.²⁶⁻²⁸ **STE represents a**
**photoinduced electron-hole pair (exciton) mediated by the interaction between exciton**
**and corresponding lattice.** The formation of localized STE is critically dependent on
the dimensionality of the crystalline systems, and lowering the dimensionality makes
exciton self-trapping easier.²⁹ Therefore, 0D systems in our study with strong
confinement are reasonably expected to be favorable for the formation of STEs. To
better understand the pressure induced emission in Cs₄PbBr₆ NCs, the in situ
high-pressure angle dispersive synchrotron X-ray diffraction (ADXRD) patterns of
the Cs₄PbBr₆ NCs were collected as shown in Figure S4 and Figure S5. The evolution
of ADXRD diffraction patterns demonstrates that a reversible structural phase
transition from rhombohedral (Phase I) to monoclinic (Phase II) structure occurs at

3.04 GPa and ends at 4.01 GPa, ~~accompanied by a significant distortion of $[\text{PbBr}_6]^{4-}$~~
~~octahedra to accommodate the Jahn-Teller effect (Figure S9).~~ First-principle enthalpy
calculations indicate that the monoclinic structure is energetically more favorable than
the rhombohedral phase with increasing pressure (Figure S7). The phase transition
accompanied by a significant distortion of $[\text{PbBr}_6]^{4-}$ octahedra (Figure S8). In addition,
the evolution of Raman spectra ranging from 70 to 300 cm^{-1} was shown in Figure S9.
Three lattice modes at 77, 88 and 127 cm^{-1} are observed at ambient conditions. These
strong modes are associated with the vibrational modes of $[\text{PbBr}_6]^{4-}$ octahedra.³⁰ As
the pressure increased to 3.10 GPa, the two lattice modes in the 70-100 cm^{-1} region
become so weak, while the relative intensity of lattice mode at 127 cm^{-1} is enhanced
dramatically, consistent with the result of ADXRD. The three modes undergo a
redistribution of intensities and remain stable beyond 4.08 GPa, reflecting that the
structure of octahedra happened to be changed due to the phase transition. This
phenomenon is also well agreed with the result of Rietveld refinements. ~~In this OD~~
~~perovskite materials, the Cs^+ cations isolate each $[\text{PbBr}_6]^{4-}$ octahedra. Thus, the~~
~~Cs_4PbBr_6 is considered to exhibit the intrinsic properties of the individual $[\text{PbBr}_6]^{4-}$~~
~~octahedra, and the photoluminescent properties are explained to be not as a result of~~
~~lattice defects but rather due to excited state structural reorganization within~~
~~individual $[\text{PbBr}_6]^{4-}$ octahedra.^{18,28} Such excited state structural reorganization in this~~
~~OD perovskites is related with the exciton self-trapping.²⁸ Therefore, the appearance~~
~~of the emission at high pressure is possibly related with the structural change of~~
~~octahedra within monoclinic phase. Herein, pressure may be an important driving~~
~~force to increase the structure distortion and thus promoting the radiative~~
~~recombination of self-trapped exciton. ~~The radiative recombination of STEs gives rise~~~~
~~to a class of mid-band-gap emission associated with the Jahn-Teller distortion. Hence,~~
~~the high-pressure emission has nothing to do with the variation of the intrinsic band~~
~~gaps.~~ Note that as the pressure increases in the stability field of the initial
rhombohedral phase, the strong electrostatic interaction makes the six bond lengths of
Pb-Br smaller within the regular $[\text{PbBr}_6]^{4-}$ octahedra (Figure S10). During this process

1 the octahedra do not distort (Table S1), and thus no PL response is seen below 3.01
2 GPa.

When the pressure increases beyond roughly 6.23 GPa, the PL intensity of the
compressed samples decreases with increasing pressure, and almost disappears at
18.23 GPa (Figure 2c). The weakening of PL should be ascribed to the deviatoric
stress arising from nonhydrostatic conditions in the diamond anvil cell (DAC), which
was indeed reported from earlier studies using synchrotron radiation small-angle x-ray
scattering (SAXS).³¹⁻³³ Deviatoric stresses eventually lead to sluggish amorphization
of Cs₄PbBr₆ NCs upon further compression, which was largely related to the higher
degree of distortion and random orientations of inorganic octahedra within the
material.^{25,34} Recovered sample was also characterized by TEM, as shown in Figure
S11. We can see that the quenched samples exhibit an aggregation, to a large extent,
after the high-pressure treatment. It appears, therefore, that deviatoric stress should be
a crucial factor in reducing the PL intensity. We further investigated the PL responses
with pressure through a control high-pressure experiment by directly dispersing the
samples in toluene, and then loading it into the DAC (Figure S12).^{11,13} It was found
that the samples dispersed in toluene still exhibited a fluorescence enhancement,
followed by persistent decrease in PL intensity. Note that the enhanced emission
process could be maintained up to pressures of 7.89 GPa, much higher than that in
experiment by adopting silicon oil as pressure transmitting medium (PTM).
Deviatoric stress also appears to exist even when the sample was loaded into the DAC
in an aggregation form. The optical images (Figure S13) clearly demonstrate the
bright trend of PL brightness in Cs₄PbBr₆ NCs with pressure, accompanied by the
color changes from bright white to dark yellow. Furthermore, we recorded the
chromaticity coordinates of emission upon compression from 3.01 to 18.23 GPa
(Figure 2d and Table S2). Based on the CIE chromaticity diagram, the emission at
3.01 GPa lies to the yellow side of pure white light (0.32, 0.32) with a corresponding
color temperature (CCT) in the range of 3000-4500 K, resulting in the “warm” white
light for many indoor lighting applications. In addition, the luminescence colors of the
resulting Cs₄PbBr₆ NCs changes from white to dark yellow as the pressure increases

from 3.01 GPa to 18.23 GPa. Therefore, we indeed developed high-pressure
technology as a robust tool to achieve not only PIE, but also to tune the chromaticity
of emission. The color tunability of 0D Cs₄PbBr₆ NCs allows for the development of
optically-pumped WLEDs with different photometric properties for various
applications, such as navigation lights for airplanes and military signs, where lighting
specifications need to be adjusted on demand.

**2.3 In situ high-pressure absorption spectra of Cs₄PbBr₆ NCs**

**Revised Figure 3.** (a) Typical profile of absorption band against pressures for Cs₄PbBr₆ NCs in
situ measured in a DAC apparatus. The dashed arrows indicate the shift of bound exciton
absorption peaks of E_{ex1} and E_{ex2}. (b) Absorption spectra of Cs₄PbBr₆ NCs measured at a selected
pressure of 6.17 GPa. Therein, E_{ex1}, and E_{ex2} represent the splitting bound exciton absorption
peaks. (c) Pressure dependence of energy difference ΔE deduced from the difference of E_{ex1} and
E_{ex2}.

To elucidate the origins of the observed PIE phenomenon in Cs₄PbBr₆ NCs, the
pressure-dependent absorption of Cs₄PbBr₆ NCs was determined (Figure 3a). We find
that the absorption peak experiences a slight redshift with increasing pressure. When
the pressure increased beyond 3.02 GPa, the profile of the absorption peak underwent
a stark change at the same time as the sudden appearance of broad band emission.
Both the distinct changes in absorption and PL spectra at 3.02 GPa are associated with

a structural phase transformation in Cs₄PbBr₆ NCs seen at high pressure. Moreover,
an additional shoulder at the low-energy side of the original absorption band appeared
at 3.53 GPa (Figure 3a). As pressure increases to 6.17 GPa, this shoulder develops
into a clearly-resolved sharp peak with an increased magnitude, and remains stable up
to 18.14 GPa. We attributed the formation of two absorption peaks to the splitting of
the excitonic peak. As the absorption bands of perovskites are greatly influenced by
the nature of the octahedra. Therefore, the formation of these two absorption peaks
should be highly related to the distortion of [PbBr₆]⁴⁻ octahedra. As shown in Table
S1, although the bond lengths of Pb-Br decreased with compression, the [PbBr₆]⁴⁻
octahedra exhibited no any distorted configuration below 3.01 GPa. However, in
high-pressure monoclinic phase, the three Pb-Br bonds happen to be non-equivalent
within [PbBr₆]⁴⁻ octahedra persistent change upon compression, forming two classes
of Pb-Br bonds. Therein, the breaking symmetry of the [PbBr₆]⁴⁻ unit in the ground
state under pressure was very likely lead to the splitting of the excitonic peak in two:
one at higher and one at lower energies. This assignment is further supported by the
absorption oscillator strengthes using the excited-state structure associated with
self-trapped exciton (Figure S14) at the single-particle level. We found that in the
lower energy region the high-pressure phase shows two separated large oscillator
strengthes, implying the signature of peak splitting, while the ambient-pressure phase
shows one group of oscillator strengthes. Likewise, the energy difference ΔE between
the two exciton absorption peaks (Figure 3b) can also indicate the extent of distortion
for octahedra with increasing pressure. As illustrated in Figure 3c, we find that this
energy difference sharply increases in the pressure region from 4.03 to 6.17 GPa with
a pressure coefficient of 0.08 eV/GPa (Figure 3c). When the pressure increases
beyond 6.17 GPa, the ΔE experienced a relatively sluggish increase. Figure 3c
demonstrated that the distorted extent of octahedra was improved with compression,
but beyond 6.17 GPa, the [PbBr₆]⁴⁻ octahedra started to tilt and undergo a
considerable rotation deviating from the original orientation in disordered way,
consistent with the result of high-pressure ADXRD.

Moreover, since high pressure causes the distortion of the [PbBr₆]⁴⁻ octahedron,

the excited state structural reorganization required to trap a photoexcited exciton is
expected to be smaller than that in the ambient-pressure phase containing perfect
$[\text{PbBr}_6]^{4-}$ octahedra. The resulted smaller Stokes shift may allow reasonable
wavefunction overlap between the excited state (associated with self-trapped exciton)
and the ground state, leading to a likely non-zero or enhanced transition dipole
moment compared to the ambient-pressure phase. Given that, we calculated the
transition dipole moments based on the excited-state structure associated with
self-trapped exciton at the single-particle level, as shown in Figure S14. We found
that the high-pressure phase exhibits a one-magnitude larger oscillator strength (~ 0.06
a.u.) at the lowest excitation energy than that of the ambient-pressure phase (~ 0.003
a.u.) This implies that the pressure-induced distortion of the $[\text{PbBr}_6]^{4-}$ octahedra
promotes the self-trapping excitonic state to be more optically active.

In addition to some extent enhanced optical activity, our further first-principle
calculations show that the self-trapped exciton in the high-pressure phase has the
larger exciton binding energy (1.31 eV) than that of the ambient-pressure phase (1.13
16 eV). This originates from the increased electron-phonon coupling strength when the
17 $[\text{PbBr}_6]^{4-}$ octahedra are contracted upon compression, accompanying with the
18 strengthened Pb-Br covalent bondings (see Table S1). Since the self-trapped exciton
is mediated by the interaction between exciton and lattice distortion, the stronger
electron-phonon coupling can more effectively bind photoexcited carriers to form
self-trapped exciton in the lattice distortion field. The Figure S15 shows the
comparison of electronic band structures with and without the lattice distortion
mediating self-trapped exciton for the ambient-pressure (1 atm) and high-pressure (4
24 GPa), respectively. As seen, the electronic band structure is much more seriously
distorted upon the presence of lattice distortion for the high-pressure phase, which
ambiguously indicates the stronger electron-phonon coupling. Such increased
electron-phonon coupling strength is responsible for the larger binding energy of
self-trapped exciton in the high-pressure phase.

1

2 **Figure 4.** Configuration coordinate model of emission for the Cs_4PbBr_6 NCs at 1 atm (a) and 4
 3 GPa (b). Therein, the absorption transition upon excitation is described from A to B. The
 4 self-trapped exciton state (STE) recombination emission is depicted from C to D. The path
 between B and C refers to exciton self-trapping (red) and detrapping (green). (E_{detrapp}): activation
 energy for detrapping, (E_{ex1} and E_{ex2}): the splitting of bound exciton state, (ST): self-trapped state,
 (G): ground state, $S^{1/2}$: Huang-Rhys parameter.

Figure 4 depicts a schematic of the emission processes in Cs_4PbBr_6 NCs with
 and without compression. The excitation transition is described from A to B. Due to
 the unique structure of Cs_4PbBr_6 NCs, the excited carriers are readily localized to
 form bound excitons from the conduction band due to strong quantum confinement.
 The formed bound excitons subsequently relax to the self-trapped state via B to C. At
 ambient condition, the ideal octahedral structures have low activation energy for
 detrapping due to the weak electron-phonon coupling strength. Therefore, the
 photoexcited carriers are readily detrapped from self-trapped state to bound exciton
 state by thermal activation. In addition, the low optical activity as we mentioned
 above also hinders the appearance of emission. However, at 4 GPa, the distorted
 octahedra within high-pressure monoclinic Cs_4PbBr_6 NCs result in a decrease in
 Stokes shift, which allow reasonable wavefunction overlap between the excited state
 (associated with self-trapped exciton) and the ground state. This lead to an enhanced

optical activity compared to the ambient-pressure phase. In addition, the distorted
octahedral structure processes a stronger electron-phonon coupling strength (the
Huang-Rhys parameter $S^{1/2}$ can reflect the strength of electron-phonon coupling since
$S^{1/2}$ generally increases with the increase in electron-phonon coupling strength),
resulting in an enhancement of the activation energy for detrapping and thus the STEs
can hardly convert to bound excitons by thermal activation. Subsequently, there
appears the broad emission associated with the transition of the STEs to the valence
band through the radiative recombination, as illustrated from C to D (Figure 4b). With
much higher pressure, the $[\text{PbBr}_6]^{4-}$ octahedra are further contracted and the Pb-Br
covalent bondings become more strengthened (Table S1), which further enhanced the
electron-phonon coupling strength. This enables the increase in the concentration of
STEs, thus improving the possibility of radiative recombination. Therefore, the
persistently increase in PL intensity can be observed with compression.

**3. Conclusion**

In summary, we have investigated the structural and optical response of the 0D
all-inorganic MHP, Cs_4PbBr_6 NCs, as a function of pressure up to 18 GPa.
Intriguingly, a pressure-induced emission (PIE) and a subsequent large emission
enhancement behavior were achieved upon high-pressure processing. The synchrotron
ADXRD data and Rietveld refinement results strongly indicated that the unexpected
emission is associated with the deformation of $[\text{PbBr}_6]^{4-}$ octahedra in high-pressure
monoclinic phase, which is also supported by first-principle energetic calculations.
We ascribe the PIE to radiative recombination of the self-trapped excitons, which are
emerged associated with the large distortion of Pb-Br octahedra after phase transition.
**Joint experimental and theoretical analyses indicate that the emergence of room**
**temperature luminescence may be ascribed to the enhanced optical activity and the**
**increased binding energy of self-trapping excitonic states under high pressure.** Our
findings not only provided the fundamental relationship between structural variations
and optoelectronic properties of Cs_4PbBr_6 NCs, but also offered insight into the
microscopic physiochemical mechanism of the MHP nanosystems at extremes.

4. Experimental Section

[revised manuscript text omitted]

$\text{Cs}_4\text{PbBr}_6/\text{CsPbBr}_3$. *J. Phys. Chem. Lett.* **9**, 830-836, (2018).
- 39 Kresse, G. & Furthmüller, J, Effect of Er doping on the electronic structure optical properties
of ZnO. *Phys. Rev. B*, **54**, 11169-11186, (1996).

Reviewers' Comments:

Reviewer #3:

Remarks to the Author:

I am grateful the authors took into account my criticism. I now find the manuscript in a format suitable for publications.